

# Scanning electron microscopy (SEM) reveals high diversity of setae on the hind tibiae and basitarsi of Peruvian Stingless Bees (Apidae: Meliponini)

Marilena Marconi[1], Carlos Daniel Vecco-Giove[2], Javier Ormeño Luna[2], Agustín Cerna Mendoza[2], Emiliano Mancini[3] and Andrea Di Giulio[4]

[1] Department of Environmental Biology, University of Roma "La Sapienza", Rome, Italy
[2] Faculty of Agricultural Sciences, Universidad Nacional de San Martín, Tarapoto, Peru
[3] Department of Biology and Biotechnology "Charles Darwin", University of Roma "La Sapienza", Rome, Italy
[4] Department of Science, Roma Tre University, Rome, Italy

## ABSTRACT

Stingless bees belong to the group of corbiculate bees, all characterized by the presence of a corbicula, a specialized structure of the hind tibia used for pollen collection. This group exhibits significant variation in foraging behavior, from flower-visiting foragers to kleptoparasites and obligate necrophagous. So far, scanning electron microscopy (SEM) studies have been mainly focused on the hind leg setae of *Apis* spp. and *Bombus* spp. Here, we performed a comparative morphological analysis of the setae and the pollen handling structures in the hind tibiae and basitarsi of seven stingless bee species: the floral pollen collector bees *Melipona* cf. *eburnea*, *Partamona testacea*, *Scaura* cf. *latitarsis*, *Trigonisca* cf. *atomaria*, *Trigona dallatorreana*, the robber bee *Lestrimelitta* sp. and the obligate necrophagous *Trigona* cf. *hypogea*, collected in Peruvian forests in 2020. The setae were classified into simple and branched types, subdivided into five and seven subtypes, respectively. In addition, we described three types of penicillum, five types of rastellum, three types of pollen brush, two types of setae covering the auricular area, and show the setae forming the sericeous area of *Trigona* spp. Finally, seven types of keirotrichia are described. We highlight that certain types of setae show a high degree of conservation across species, while others are unique and potentially indicative of adaptive specialization. Among species that have abandoned pollen foraging on flowers, we observed a simplification in the number of setal types and the absence of pollen handling structures in *Lestrimelitta* sp., whereas *T.* cf. *hypogea* exhibits the highest diversity of setae and retains most of these structures. Finally, based on these two species, we reflect on the link between reduced corbicula and functional corbicula. The study highlights the importance of further exploring the mechanical and sensory roles of setae and expanding this research in Meliponini. Such investigations can improve our understanding of their adaptive functions and provide valuable insights into the functional ecology, taxonomy and phylogenetic studies of these tropical pollinators.

Corresponding authors
Marilena Marconi,
marilena.marconi@uniroma1.it
Andrea Di Giulio,
andrea.digiulio@uniroma3.it

## INTRODUCTION

Bees primarily forage on flowers to collect pollen, which serves as the main food for their offspring. This feeding habit has led to the evolution of various specialized structures that enable bees to collect and transport pollen to the nest. Some examples of these structural adaptations include the hairiness of their bodies (*Stavert et al., 2016*) and the brushes of plumose setae on their hind legs and underside of the abdomen, known as scopae (*Michener, 1944*). On the other hand, the females of the corbiculate bees (Apini: *Apis* Linnaeus 1758; Meliponini: 59 genera; Bombini: *Bombus* Latreille 1802; Euglossini: five genera) (*Ascher & Pickering, 2020*) possess the most specialised structure to carry up the pollen: the corbicula. The distal part of their hind tibiae, in fact, enlarges into a bare and often concave area, known as the pollen basket (*Michener, 1999*), where the pollen grains are compacted into a pellet, and transported to the nest. In addition to corbicula, these bees possess other specialised structures on their hind tibiae and basitarsi, such as setae forming combs and brushes (*e.g.*, penicillum, rastellum and the pollen brush), which play key roles in handling and loading pollen into the corbicula (*Michener, 2007*).

The term '*seta*' originates from Latin (*sēta*, *saeta*) and is defined by the Oxford Dictionary of Zoology as a 'stiff, hair-like or bristle-like structure in many invertebrates' (*Oxford Reference, 2020*). It is a cuticular projection found in arthropods, often serving both chemosensory and mechanosensory functions (*Garm, 2004*; *Winterton, 2009*). Beyond their mechanoreceptive function, insect setae serve various other roles, contributing to camouflage, defence, pheromone dispersal, sexual display and grooming (*Jander, 1976*; *Peters et al., 2016*; *Williams, 2007*; *Winterton, 2009*). The terminology used to describe setae has historically been inconsistent, with researchers employing a variety of often interchangeable terms such as '*hairs*', '*bristles*', '*trichiae*', '*aculei*', and '*chetae*' (*Winterton, 2009*). In the context of bees, both hairs and bristles are commonly used (*Müller & Kuhlmann, 2003*; *Hines et al., 2022*; *Khan & Liu, 2022*). In this work, to ensure terminological consistency and avoid ambiguity, we adopt the term 'seta' as a general descriptor encompassing these structures.

In bees, the setae are essential for pollen collection (*Thorp, 1979*). Those located around the corbicula of *Apis mellifera* Linneus 1758 and *Apis dorsata* Fabricius 1793, primarily fulfill a mechanosensory role. In fact, they respond to variations in the amount of pollen gathered, allowing for dynamic adjustment of the load and ensuring its stable retention during flight (*Ford et al., 1981*). More broadly, across bee taxa, setal morphologies have been associated with foraging ecology as distinct setal types are linked to the collection of pollen with various characteristics, such as size, stickiness, and sculpturing (*Thorp, 2000*; *Konzmann et al., 2023*). Generally, in corbiculate bees there are two types of pollen packaging. In Type I, the pollen is placed directly into the corbicula by the middle legs, without the involvement of combs and brushes of the hind tibiae and basitarsi (*Michener, Winston & Jander, 1978*). Type II, can be either (a) ipsilateral, in which pollen is placed outside the tibia-tarsus joint by the middle leg of the same side, and then pushed into the corbicula; (b) contralateral, in which pollen is transferred from the middle leg to the pollen brushes of the hind basitarsi, then the hind legs are quickly rubbed against each

other in a pumping motion, so that the rastellum of each hind tibia scrapes pollen from the pollen brush of the contralateral hind basitarsus (*Michener, Winston & Jander, 1978*). Presumably such loading movements are derived from similar grooming movements, and the contralateral Type II is considered the most advanced (*Michener, Winston & Jander, 1978*).

Despite the high specialization of the hind legs of corbiculate bees, extensive scanning electron microscopy (SEM) studies have focused only on the hind leg setae in Apini (*Apis mellifera* Linnaeus 1758 and *Apis dorsata* Fabricius 1793) (*Casteel, 1912*; *Hodges, 1952*; *Michener, Winston & Jander, 1978*; *Southwick, 1985*; *Conde-Boytel, Erickson & Carlson, 1989*; *Saha, Ghorai & Bera, 2004*; *Khan & Liu, 2022*) and Bombini (*Bombus* spp.) (*Hines et al., 2022*), with only a few species of Meliponini receiving attention: *Friesella schrottkyii* Friese 1900, *Partamona cupira* Smith 1863, *Scaptotrigona xanthotrica* Moure 1950, *Tetragonisca angustula* Latreille 1811 (*Alves, Patricio & Soares, 2003*), *Geniotrigona lacteifasciata* Cameron 1902, *Tetragonula melanocephala* Gribodo 1893, and *Tetragonula sirindhornae* Michener & Boongird 2004 (*Zubaidah et al., 2017*). However, these studies on Meliponini provide SEM images of setae from only a few regions of the hind tibiae and basitarsi, and do not offer a comprehensive description or categorization of the setal types. As a result, setal diversity remains underrepresented and largely unexplored in this group.

The stingless bees (tribe Meliponini), with 605 species distributed in tropical and subtropical regions of the world, are the most diverse lineage of corbiculate bees (*Engel et al., 2023*). This group shows considerable variation in size, hairiness, form and behaviour (*Grüter, 2020*). The sizes range from the most primitive and tiny (2–4 mm body length), as in *Trigonisca* Moure 1950, to the largest, robust, and hairy (8–15 mm), such as in *Melipona* Illiger 1806 (*Grüter, 2020*). They also differ in their foraging behaviour, ranging from floral pollen collectors to cleptoparasites (*Lestrimelitta* Friese 1903 and *Cleptotrigona* Moure 1961 genera) (*Eardley, 2004*; *Gonzalez & Griswold, 2012*) stealing pollen from the nests of other stingless bees, or the obligate necrophagous (*i.e., Trigona hypogea* group) that feed on carrion (*Camargo & Roubik, 1991*).

The hind legs of Meliponini exhibit remarkable structural adaptations that support a wide range of functions across the tribe. While pollen and resin transport, as well as wing grooming (*Michener, 2007*), are widespread behaviors among stingless bees, other tasks—such as carrying seeds, mud and feces for nest construction—are restricted to some species (*Grüter, 2020*). Notably, the penicillum—a brush-like structure composed of setae on the hind tibiae—is a unique feature of Meliponini (*Engel & Rasmussen, 2020*). It functions to move pollen upward into the corbicula during tibia–basitarsus articulation performed in Type II movements (*Michener, 2007*). Furthermore, the rastellum—a comb of robust setae arising from the inner distal margin of the tibia—serves to rake pollen from the pollen brush of the contralateral hind basitarsus during Type II movement (*Michener, 2007*). Finally, in the hind basitarsus, there is an auricular area, bordered by a fimbriated line of setae, presumably with a similar function to the auricle, or pollen press, of *A. mellifera* (*Michener, Winston & Jander, 1978*). In stingless bee *Scaura latitarsis* Friese 1900, the hind basitarsus developed a bladder-like structure, a specialized adaptation that may assist foragers in collecting pollen that has fallen to the ground (*Roubik, 2018*).

Among highly eusocial bees, the complete abandonment of pollen collection from flowers and the adoption of alternative foraging behaviors, such as kleptoparasitism and necrophagy, occur exclusively in Meliponini (*Barbosa Noll et al., 1996*). This has resulted in distinct morphological adaptations of the hind legs in some species (*Grüter, 2020*). Robber bee workers of the genus *Lestrimelitta* exhibit reduced or absent corbiculae and lack the penicillum (*Friese, 1931*; *Parizotto, 2010*), although *Lestrimelitta limao* Smith 1863 can still use their hind tibiae to transport nesting materials or pollen raided from other bee colonies (*Zuben & Nunes, 2014*). Similarly, carrion-foraging workers of *Trigona hypogea* Silvestri 1902 possess a reduced and flattened corbicula, which is still functional for collecting resin (*Roubik, 1982a*; *Roubik, 1989*). This behavioral diversity within Meliponini offers a valuable opportunity to investigate how specialized structures involved in pollen handling, such as the setae on the hind tibiae and basitarsi, are conserved, modified, reduced or lost in response to different ecological pressures associated with divergent foraging strategies.

The aim of this study is to highlight the diversity of the setal types and pollen handling structures in stingless bees by SEM analysis of the hind tibiae and basitarsi of floral pollen collector species *Melipona* cf. *eburnea* Friese 1900, *Partamona testacea* Klug 1807, *Scaura* cf. *latitarsis* Friese 1900, *Trigonisca* cf. *atomaria* Cockerell, 1917, *Trigona dallatorreana* Friese 1900, the robber bee *Lestrimelitta* sp. and the obbligate necrophagous *Trigona* cf. *hypogea* Silvestri 1902. Specifically, we describe and categorize setal types, map their distribution on the hind tibiae and basitarsi, and examine structural variation among these species. In particular, we identify which setal types are most conserved and explore how they vary in floral pollen-collecting bees and those that have secondarily abandoned this behavior through the adoption of kleptoparasitic or necrophagous foraging strategies. Finally, we addressed the meaning of 'functional corbicula' and emphasised the importance of extending research on the setae of other stingless bee species for taxonomic, phylogenetic and functional ecology studies.

## MATERIALS & METHODS

### Material examined

Here, we examine the morphology of the hind tibiae and basitarsi and the diversity of the setal types in seven worker stingless bees, each representing a different species (Table 1). The samples were collected in 2020 from Peruvian forests (see *Marconi et al., 2022*), except for the *Lestrimelitta* sp. sample, which was collected during a separate survey in the Tingo Maria National Park (Peru), in the same year. Specimens attracted to the bait stations were captured using fine-mesh nets to minimize potential damage to the setae and were preserved in Falcon tubes containing 96% ethanol.

All the samples were collected with permits issued by the Servicio Nacional Forestal y de Fauna Silvestre of Peru (N° 60-2016 SERFOR/DE) and the specimen vouchers deposited at the Estudios Amazonicos Biological Material Depositary Center (Tarapoto, Peru).

### Scanning electron microscopy (SEM)

For microscopy, the third leg of each of the seven specimens was cut off at the proximal segment, rinsed twice in 100% ethanol to ensure complete dehydration, and then subjected

**Table 1  Stingless bee species examined in this study.** For each specimen, the locations, coordinates (WGS84), and altitude of the sampling sites are given.

| Sample code | Species | Locality (Department) | Coordinates (WGS84) | | Altitude (m a.s.l.) |
|---|---|---|---|---|---|
| | | | **N** | **E** | |
| TM1 | *Lestrimelitta* sp. Friese, 1903 | Tingo Maria National Park (Huánuco) | −9°21′20.5776″ | −76°2′12.6672″ | 1,095 |
| PY23 | *Melipona* cf. *eburnea* Friese, 1900 | Pabloyacu (San Martín) | −6°04′6.3984″ | −76°56′24.8388″ | 1,200 |
| JPA003 | *Partamona testacea* Klug, 1807 | Juliampampa (San Martín) | −6°26′3.5556″ | −76°19′47.5896″ | 1,100 |
| JPA004 | *Scaura* cf. *latitarsis* Friese, 1900 | Juliampampa (San Martín) | −6°26′3.5556″ | −76°19′47.5896″ | 1,100 |
| PY8 | *Trigona dallatorreana* Friese, 1900 | Pabloyacu (San Martín) | −6°04′6.3984″ | −76°56′24.8388″ | 1,200 |
| PY4 | *Trigona* cf. *hypogea* Silvestri, 1902 | Pabloyacu (San Martín) | −6°04′6.3984″ | −76°56′24.8388″ | 1,200 |
| MA9-1 | *Trigonisca* cf. *atomaria* Cockerell, 1917 | Mangamanguilla (Piura) | −5°18′46.5228″ | −79°51′51.0840″ | 140 |

to critical point dried (BalTtec CPD 030). Subsequently, the legs were mounted on aluminum stubs by self-adhesive carbon disks, gold sputtered (Emitech K550) and analysed by a Zeiss Sigma Gemini 300 FE-SEM at the Laboratorio Interdipartimentale di Microscopia Elettronica (L.I.M.E.) of Roma Tre University (Rome, Italy).

## Morphology of the hind tibiae, basitarsi and setae

For each specimen, the length of the hind tibia (TL) was measured from its proximal articulation with the femur to the distal margin, while the length of the basitarsus (BL) was measured from its proximal articulation with the tibia to its distal end. The corresponding breadths of the tibia (TW) and basitarsus (BW) were measured at their widest points (Fig. S1) (*El-Aw et al., 2012*; *Haldhar et al., 2021*; *Siraj et al., 2022*).

For setae types that exhibit variation in length of the shaft, a range is provided, indicating the shortest and longest observed. When no appreciable variation in length was observed, mean length and standard deviation (SD) were calculated based on three representative setae measured from a single specimen.

This methodology also applies to the setae that make up the inferior and superior parapenicillum, the penicillum and rastellum. For branched setae, the divergence angle of the outgrowths from the shaft was visually estimated. Setae length data were not normalized to body size of the specimens and are provided for descriptive purposes only, as no morphometric analysis was conducted.

Penicillae, rastella and auricular area were compared among species in terms of the types of setae forming these structures and their degree of development, evaluated qualitatively based on the extent and continuity of their distribution. These structures were considered well-developed when setae covered a large portion of the surface and formed a continuous arrangement.

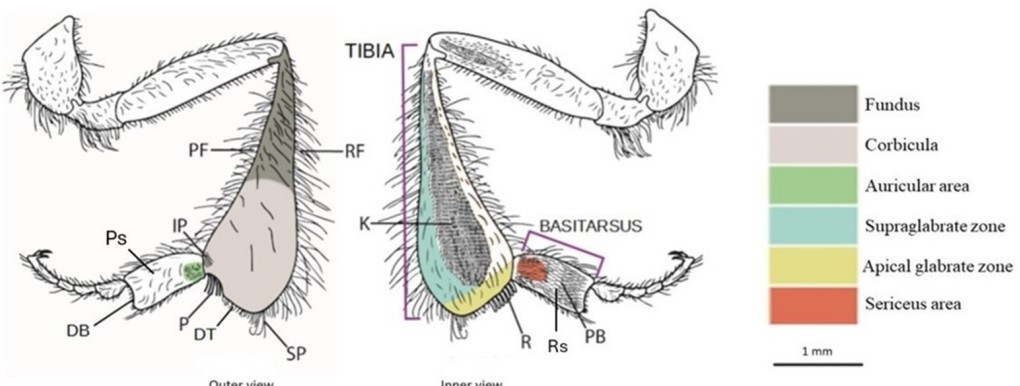

**Figure 1  Drawing of the hind leg (outer and inner views) of a stingless bees.** PF, proventral fringe; RF, retrodorsal fringe; IP, inferior parapenicillum; P, penicillum; SP, superior parapenicillum; DT distal margin of the tibia; Ps, Prolateral surface of the basitarsus; DB, distal margin of the basitarsus; K, keirotrichiate zone; R, rastellum; Rs, Retrolateral surface of basitarsus; PB, pollen brush. (Drawing by M. Marconi).

Length measurements were obtained with ImageJ (https://imagej.net/ij/download.html) using SEM photographs. The tables with photos were designed with Liberoffice Draw (https://it.libreoffice.org/).

## Terminology

The cuticular sculpture of the tibiae and basitarsi is named according to *Harris (1979)*. The terms adopted for the surface orientation plans of the tibiae and basitarsi follow *Michener (1990)* and *Engel et al. (2021)*. The definitions of the zones of tibia and basitarsus, where the setal types are described, are provided by *Michener, Winston & Jander (1978)*, *Michener (1999)*, *Michener (2007)* and *Engel et al. (2021)* (Fig. 1). The categorisation of the setae consider setal nomenclature from multiple arthropods (*Harbach & Knight, 1980*; *Larsen, 2003*; *Garm & Watling, 2013*). The functions of structures such as inferior-, superior parapenicillum, penicillum, rastellum, auricular area, keirotrichia, pollen brush and sericeus area are listed below according to *Michener, Winston & Jander (1978)* and *Michener (2007)*.

**Apical glabrate zone:** The area on the inner surface of the tibia between the keirotrichiate zone and the distal fringe (*Engel et al., 2021*).

**Corbicula:** A more or less concave space surrounded by long setae and used to transport pollen (*Michener, 1999*).

**Distal margin of the tibia (DT):** The distal margin of the tibia (*Engel et al., 2021*).

**Fundus:** The portion of the metatibia close to the corbicula. It exhibits differences in surface sculpturing, often appearing more textured than the polished corbicula, and in pilosity, being more densely covered with surface setae (*Engel et al., 2021*).

**Inferior parapenicillum (IP):** a group of setae arising from the anterior apical angle of the hind tibia, probably to assist the action of the penicillum (*Michener, 2007*).

**Keirotrichiate zone (K):** Portion of the retrolateral surface of the tibia occupied by the keirotrichia, setae of uniform length used for cleaning the wings (*Michener, 2007*).

**Penicillum (P):** It is a compact tuft of curved setae arising near the anterior apical margin of the hind tibia, usually directed posteriorly. It is used to help move pollen upward into the corbicula during tibia-basitarsus joint movements (*Michener, 2007*).

**Proventral fringe (PF):** Ventral fringe of the tibia (*Engel et al., 2021*).

**Rastellum (R):** a comb of robust setae arising from the inner distal margin of the tibia and serving to rake pollen from the pollen brush of the contralateral hind basitarsus. Typically, the first setae are longer and bent over the adjacent setae, while the last are almost straight and short (*Michener, 2007*).

**Retrodorsal fringe (RF):** Dorsal fringe of the tibia (*Engel et al., 2021*).

**Superior parapenicillum (SP):** a group of setae arising from the posterior apical angle of the hind tibia, probably to assist the action of the penicillum (*Michener, 2007*).

**Supraglabrate zone:** The area on the inner surface of the tibia between the retromarginal fringe and the keirotrichiate zone (*Engel et al., 2021*).

**Auricular area:** It is bordered by a fimbriated line of setae, presumably with a similar function to the auricle, or pollen press, of *Apis mellifera* (*Michener, Winston & Jander, 1978*).

**Distal margin of the basitarsus (DB):** The distal margin of the basitarsus (*Engel et al., 2021*).

**Pollen brush (PB):** It consists of numerous rows of setae on the inner surface of the basitarsus involved in brushing pollen from the abdomen or rubbing with other tibial combs to transfer pollen to the corbicula (*Michener, 2007*).

**Prolateral surface of the basitarsus (Ps):** Outer surface of the basitarsus (*Engel et al., 2021*).

**Retrolateral surface of the basitarsus (Rs):** Inner surface of the basitarsus (*Engel et al., 2021*).

**Sericeus area:** It is a well-defined area of short and dense setae at the base of the inner surface of the basitarsus, whose function is still unclear (*Michener, 2007*).

## RESULTS

### Comparison of hind tibiae and basitarsi morphology among stingless bees

#### Hind tibiae

Differences among species were observed in tibial morphology (Figs. S2–S3), as well as in length and breadth (Table S1), including the flatness and concavity of the corbicula, the shape of the distal margin, the extent of the supraglabrate and apical glabrate areas, and the sculpturing of the cuticle (Table S2).

*M.* cf. *eburnea* has a broad triangular shape and a smooth concave corbicula, with a striate imbricate sculpture in the fundus; the apical glabrate zone is absent. *P. testacea* has a "spoon" shape and concave corbicula, irregular imbrications, and a broad apical glabrate zone. *S.* cf. *latitarsis* has a narrow triangular corbicula with long parallel imbrications and a wide apical glabrate zone. *T. dallatorreana* has a "hockey stick" shaped corbicula with a smooth surface, wide glabrate zones and an abrupt clivulus. *T.* cf. *atomaria* has a "drop" shape with polygonal imbrications and a slightly concave corbicula; both glabrate zones

are reduced. *Lestrimelitta* sp. and *T.* cf. *hypogea* have a "club" shape and flat corbicula with a smooth surface; in *T.* cf. *hypogea* the glabrate zones are broad and the clivulus is abrupt.

### Hind basitarsi

Variation among species was found in basitarsal morphology (Figs. S4–S5), length and breadth (Table S1), the shape of the retrodorsal (rm) and distal margins (DB), the distal angle (da) and the sculpturing of the cuticle (Table S3).

*M.* cf. *eburnea* has a broad basitarsus with a humped retrodorsal margin and an acute distal angle; the surface is partially imbricated near the proventral margin and smooth elsewhere. *P. testacea* has a broad basitarsus with a gibbous retrodorsal margin and a distinct notch along the distal margin; the surface is imbricated only near the proventral margin. *S.* cf. *latitarsis* has a "parrot beak" shaped basitarsus with an acute distal angle and a strongly curved retrodorsal margin; the cuticular sculpture is partially imbricated. *T. dallatorreana* has a similarly broad basitarsus, completely smooth, with a slightly curved retrodorsal margin and a broad sericeus area on the retrolateral surface. *T.* cf. *atomaria* has the broadest basitarsus , transitioning from imbricate to smooth, with a curved retrodorsal margin and an obtuse distal angle. *Lestrimelitta* sp. has a narrow, "stick-like" basitarsus with parallel sides and straight margins. *T.* cf. *hypogea* has a basitarsus with a slightly gibbous retrodorsal margin, acute distal angle and reduced sericeus area.

## Classification of the setae

Measurements of setae lengths are provided in Table S4.

## Simple setae

Setae without outgrowths (Fig. 2).

**Narrow setae (ns):** The setae are thin and straight, with a uniform diameter along most of their length and a pointed tip. The socket is infracuticular, allowing limited movement of the seta.

**Thorn-like setae (ts):** The setae are thorn-like in appearance, wider at the basal end and narrowing to a pointed tip. The surface features narrow grooves that begin at the basal part and converge into a longitudinal carina extending along the entire length of the seta. The setae are articulated with the cuticle in a socket area, allowing flexibility in movement.

**Hair-like setae (hs): hs1:** Thin setae with a smooth surface and a gradual tapering towards a pointed tip, giving the tip a filiform and flexuous appearance. The setae emerge at a very shallow angle from the cuticle, lying almost parallel to its surface in a distinctly prostrate position. The socket is infracuticular. **hs2:** Setae with a shaft morphology that ranges from entirely 'S'-shaped along their length to predominantly linear proximally, becoming distinctly sinuous or curved only in the distal portion. The socket is an extension of the cuticle, resulting in a supracuticular articulation of the seta. **hs3:** These setae are highly filamentous and flexible. Some emerge perpendicularly from the cuticular surface and bend abruptly at right angles; others are very long and, after emerging, bend over the cuticular surface. Finally, some emerge from the surface and gradually curve upward. All these are articulated with the cuticle in a socket area.

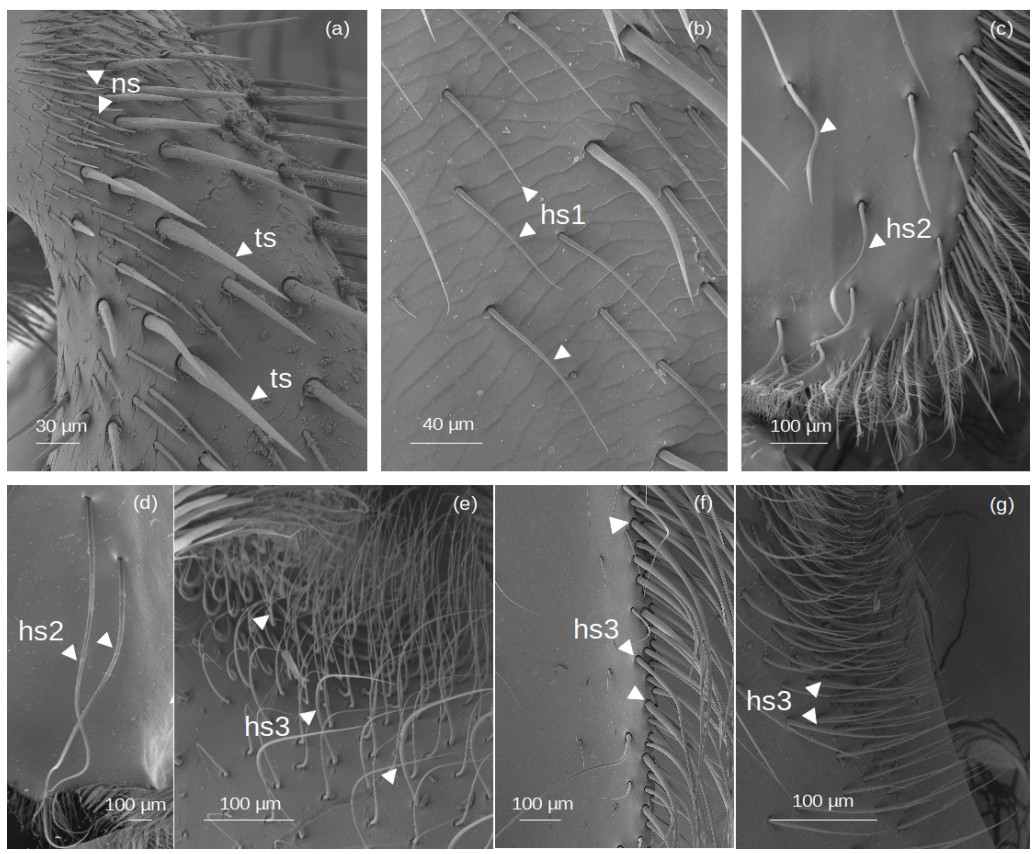

**Figure 2** **SEM microphotographs of simple setae.** (A) **ns** and **ts** types on the fundus of the hind tibia of *Lestrimelitta* sp. (B) **hs1** type on the fundus of the hind tibia of *M.* cf. *eburnea* (C) **hs2** type in the corbicula of *T.* cf. *hypogea* (D) Pair of **hs2** setae in the corbicula of *P. testacea*. (E) **hs3** type in the auricular area and (F) retrodorsal fringe in *M.* cf. *eburnea*. (G) **hs3** type on the auricular area in *P. testacea*.

## Branched setae

Setae with outgrowths originated from the setal shaft (rachis) (Fig. 3). Generally, they have a supracuticular articulation, in contrast to some dendritic setae (*i.e.,* de3; see below), where the sockets are drawn into the general cuticle.

Spinulate (sp): Setae originate from a broad basal region, gradually tapering distally to a pointed apex. Narrow grooves arise from the base and converge into a distinct longitudinal carina. Spine-like outgrowths emerge from the rachis, remaining aligned along the shaft, and in some cases, these appear in pairs, forming a double serration. They are few (3 to 5) and occur on only one side of the shaft.

Pectinate (pe): pe1: The shaft is curved, bearing thin filamentous outgrowths arranged along one side, either singly or in pairs, each diverging from the shaft at an angle of less than 45°. Towards the distal end, these filamentous outgrowths reduce in length and emerge circumferentially, giving the tip a 'pipe cleaner'-like appearance. pe2: The shaft is broad at the base and gradually narrows to a pointed tip. It can be either straight or exhibit a right-angled, elbow-like bend. Thin, elongated outgrowths arise from one side of the shaft

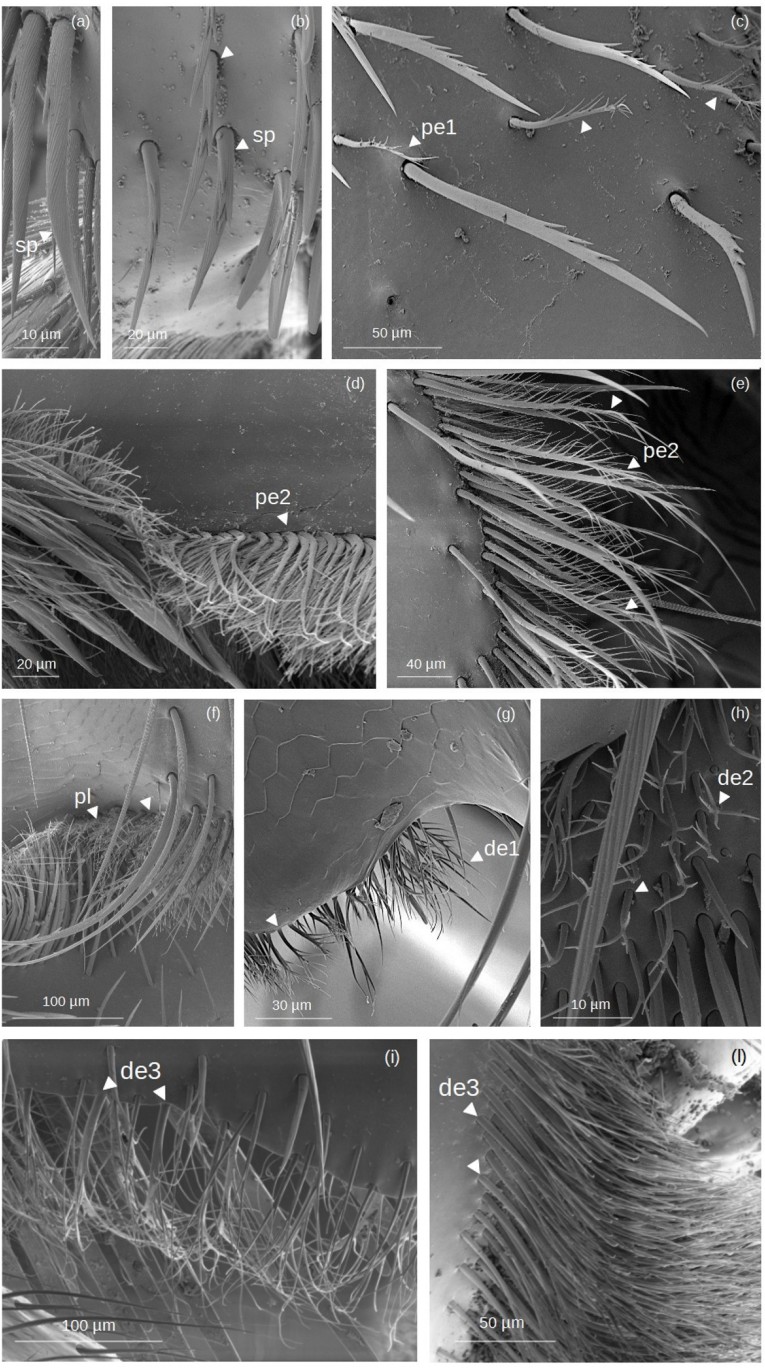

**Figure 3** **SEM microphotographs of branched setae.** (A, B) **sp** type in the proventral fringe of *Lestrimelitta* sp. and in the apical glabrate zone of the *T.* cf. *hypogea* hind tibia. (C) **pe1** type in the fundus of the hind tibia of *T.* cf. *hypogea,* (D) in the distal margin of the hind tibia of *T.* cf. *hypogea* and (E) along the retrodorsal fringe of the hind tibia of *T. dallatorreana.* (F) **pl** type along the distal margin of the hind tibia of *S.* cf. *latitarsis.* (G) **de1** type along the distal margin of the hind tibia of *T.* cf. *atomaria.* (H) **de2** type on the proximal part of the hind basitarsus (retrolateral surface) of *S.* cf. (I) **de3** type along the distal margin of the hind basitarsus of the *M.* cf. *eburnea* and (J) in the auricular area of the hind basitarsus of *T.* cf. *hypogea.*

at an angle of ~45°, extending along its entire length. One or a few short, thin outgrowths may be present randomly along the opposite side, near the tip of the shaft.

**Plumose (pl):** The shaft is long, slender, and gradually tapers to a fine, pointed distal tip. Along its entire length, it bears regularly spaced, paired lateral outgrowths, oriented bilaterally at an angle of approximately 45° toward the apex. These short, pointed projections are densely and symmetrically arranged, giving the seta a distinctly feather-like appearance.

**Dendritic (de):** The term 'dendritic' refers to a tree-like (arborescent) structure, commonly used in the description of mosquito setae (*Harbach & Knight, 1980*) and also applied to characterize the morphology of bumblebee setae (*Hines et al., 2022*). **de1:** The setae exhibit a straight shaft terminating in an acutely pointed apex. Paired, fine, acuminate outgrowths radiate from the shaft symmetrically along its two sides at an angle of ~45°, or alternatively, they may be restricted to the distal region. **de2:** The rachis is straight, oval in cross section and emerges perpendicularly from the cuticular surface. At the distal end, it may split into two or more short, pointed outgrowths.

**de3:** The rachis is thin, with a circular or ovoidal cross section, and exhibits multiple lateral divisions. These give rise to long, thin, branch-like outgrowths that are often curved and terminate in pointed or hooked apices. The branching pattern may involve an initial single lateral offshoot followed by dichotomous division at the distal end, or multiple sinuous branches arising bilaterally along the shaft. The socket is infracuticular.

**Inferior parapenicillum** (Fig. 4)

It can consist of ts, hs3, sp seta types.

**Penicillum** (Fig. 4)

**pen1:** The shaft is simple, without lateral branches, and strongly arched with an ovoid section. Dorsally convergent grooves are visible on the setae, especially near the apex, which is rounded in the more robust ones. The setae become less curved, thinner and terminate pointedly along the fascicle.

**pen2:** The shaft is not very strong, slightly sinuous with a pointed apex. Long sinuous filiform outgrowths branch off from its sides.

**pen3:** All setae in the fascicle have a strongly curved, ovoid section with a pointed apex. Sinuous filiform outgrowths branch off from them, shorter than the main stems, giving the setae a fringed appearance.

**Superior parapenicillum** (Fig. 5)

There are setae of the type hs3, pe2 and de1.

**Rastellum** (Fig. 6)

**ra1:** Setae unbranched, the first long with a broad base, tapering to a point, the adjacent ones gradually becoming more robust, flat (oval section), curved and rounded, while the last become very short.

**ra2:** Setae with spiniform outgrowths emerging often in pairs and on the same side of the stem. The first setae are long with a broad base and tapering to a point, while the other adjacent setae first become rod-shaped with a rounded tip, then flat (oval), curved and rounded, while the last setae are very short and unbranched.

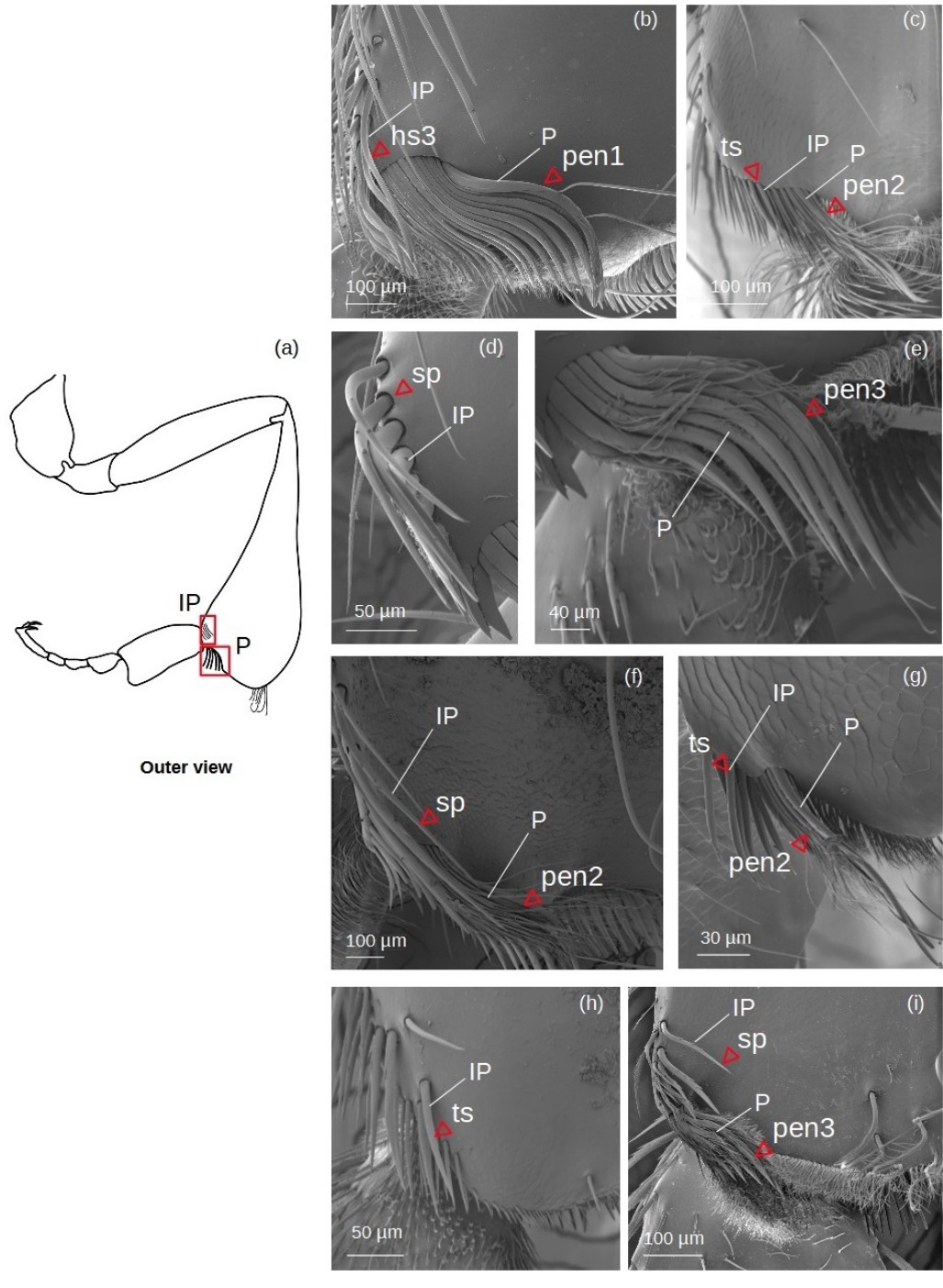

**Figure 4** **SEM microphotographs of the inferior parapenicillum (IP) and penicillum (P).** (A) Diagram showing location of the inferior parapenicillum and penicillum in the hind tibia. (B) **Hs3** setae and **pen1** form the inferior parapenicillum and penicillum of *M.* cf. *eburnea,* respectively. (C) **ts** and **pen2** types in *S.* cf. *latitarsis.* (D, E) Setae **sp** constitute the inferior parapenicillum and **pen3** type the penicillum of *T. dallatorreana.* (F) **sp** and **pen2** types in *P. testacea.* (G) **ts** setae and **pen2** types in *T.* cf. *atomaria.* (H) The inferior parapenicillum, consisting of setae **ts**, is present in *Lestrimelitta* sp., whereas the penicillum is absent. (I) In *T.* cf. *hypogea,* the parapenicillum consists of setae **sp** and the penicillum is **pen3** type.

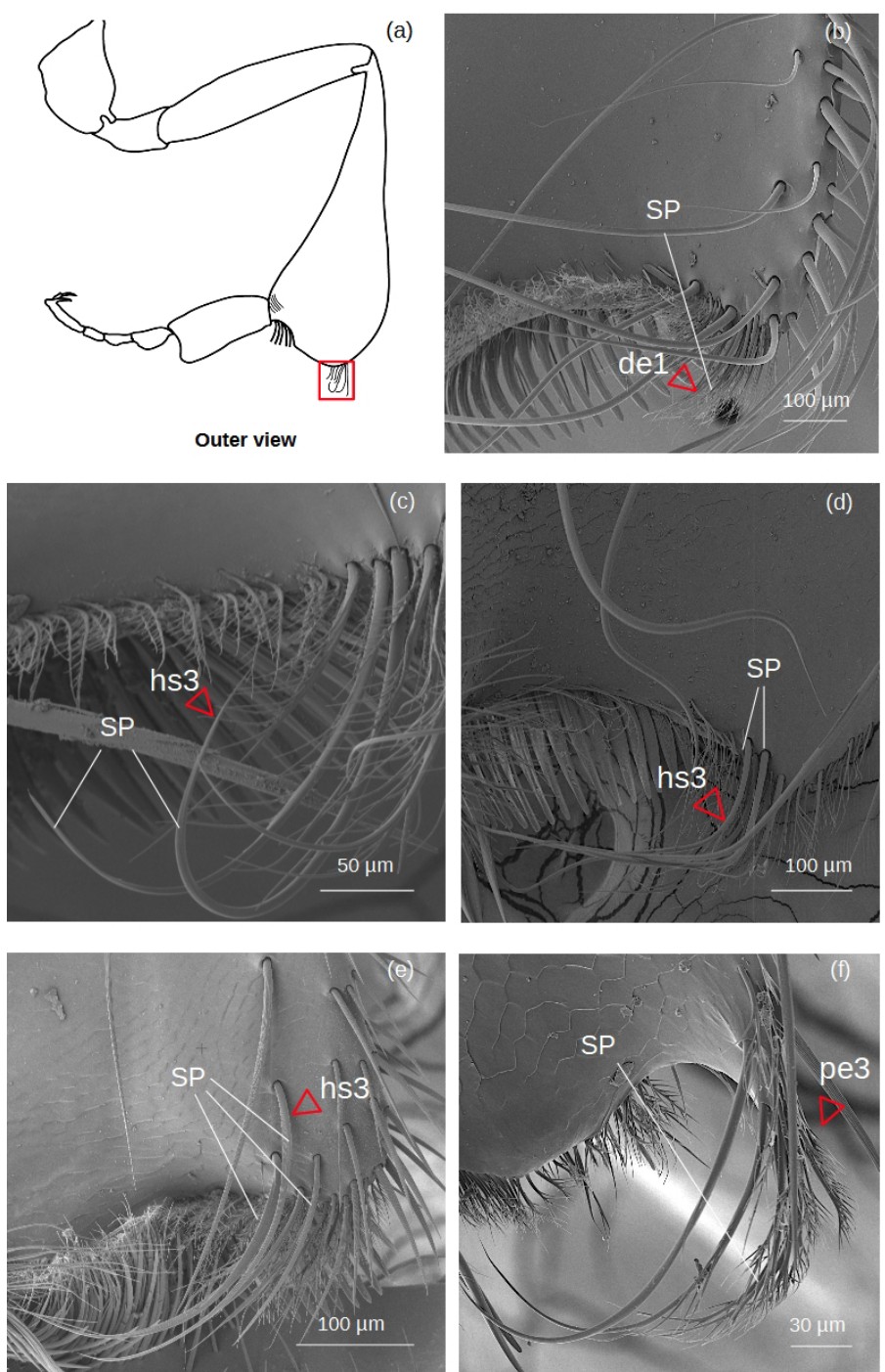

**Figure 5** **SEM microphotographs of the superior parapenicillum (SP).** (A) Diagram showing the location of the superior parapenicillum in the hind tibia. (B) **de1** type constitutes the superior parapenicillum of *M.* cf. *eburnea.* (C) Superior parapenicillum with setae **hs3** in *T. dallatorreana,* (D) *P. testacea* and (E) *S.* cf. *latitarsis.* (F) In *T.* cf. *atomaria*, the superior parapenicillum is **pe3** type. Superior parapenicillum is absent in *Lestrimelitta* sp. and *T.* cf. *hypogea.*.

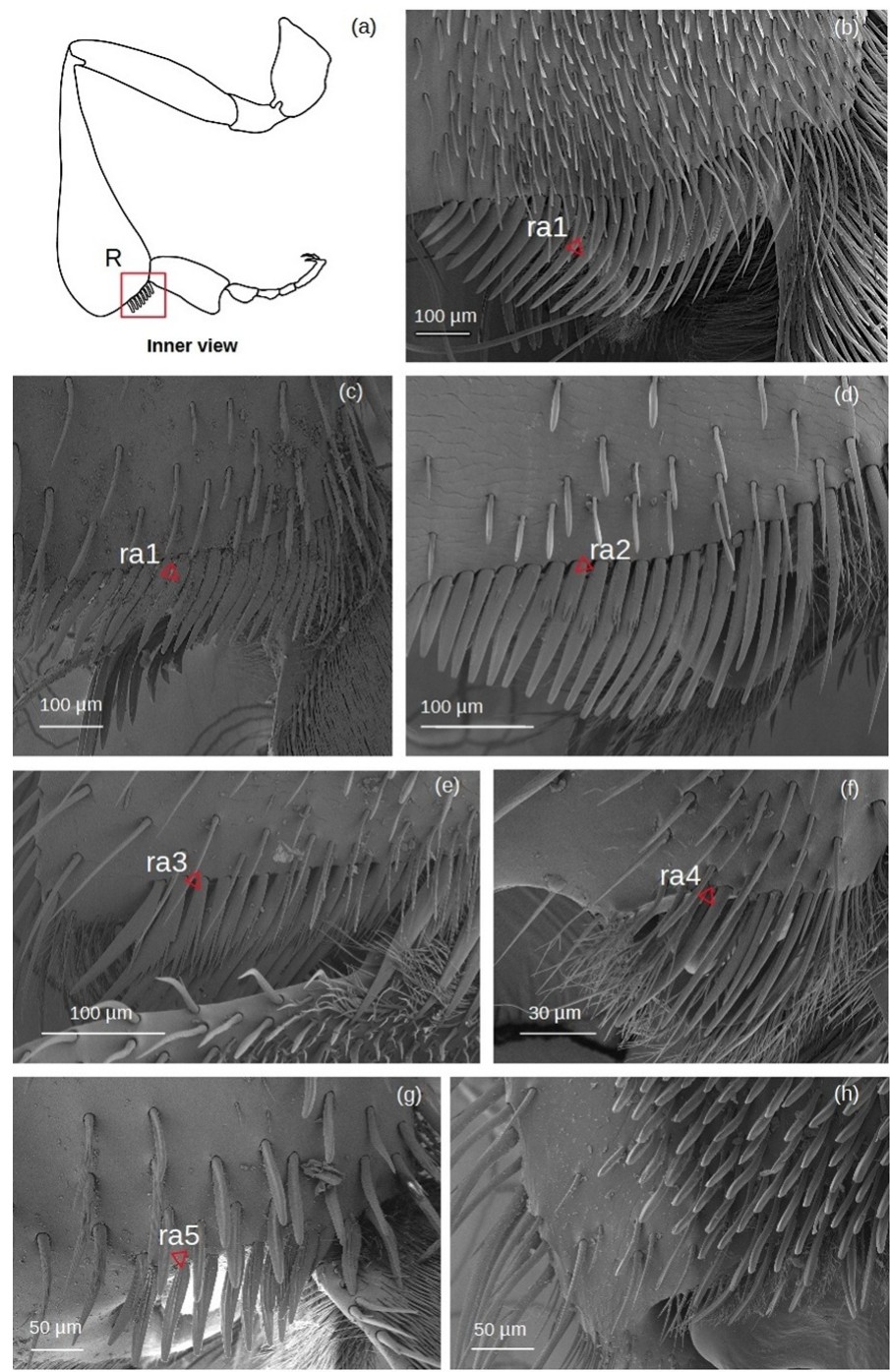

**Figure 6   SEM microphotographs of the rastellum.** (A) Diagram showing location of the rastellum in the hind tibia. The rastellum is **ra1** type in (B) *M.* cf. *eburnea* and (C) *T. dallatorreana*. (D) **Ra2** type of *P. testacea*. (E) In *S.* cf. *latitarsis* the rastellum is **ra3** type. (F) **Ra4** type in *T.* cf. *atomaria*. (G) In *T.* cf. *hypogea*, the rastellum is reduced to a few setae, type **ra5**. (H) In *Lestrimelitta* sp. the rastellum is absent.

**ra3:** Setae with numerous spiniform branches emerging successively from the same side, forming a fringe. Along the comb, the setae are all robust, curved and pointed, while the last ones are only shorter and have a rounded apex.

**ra4:** Short unbranched, slightly curved, rod-shaped setae. The former are thinner, pointed, while the adjacent ones become more robust, curved and with a rounded apex.

**ra5:** Medium sized setae without branching, strong and flat (oval section), with truncated tip, slightly curved. There are two grooves running parallel along the length of the seta.

**Pollen brush** (Fig. 7)

The sockets are an extension of the cuticle and therefore the setae have a supracuticular articulation.

**pb1:** Thin setae with rounded apex arranged in multiple, interspersed rows.

**pb2:** Tough setae with a pointed tip that emerges from the surface at an angle of approximately 45°, arranged in parallel rows.

**pb3:** Setae of type k5 arranged side by side in parallel rows.

**Setae forming the sericeus area** (Figs. 7–8)

**SA:** The setae begin with a broad circular base, widening and flattening at the apex to form a 'lancet' shape.

**Setae of the auricular area** (Fig. 9)

The types of setae that emerge from this area are hs3 and d3.

**Keirotrichia** (Fig. 10)

The setae are arranged in parallel raws, with different spacing among types.

**k1:** Thin shaft with flattened expansions emerge from the base of the shaft at its sides, running the length of it to the tip and ending in a truncated shape. Most of the setae emerge from the cuticular surface and are directed towards the distal margin of the tibia, while those on the periphery are directed towards the proventral and retrodorsal margins.

The setae on the same raw are spaced about 40 μm apart, while the rows are spaced approximately 55 μm.

**k2:** Shaft with pointed apex. The lateral expansions are wide, emerging from the base of the stem. Their edges are sinuous towards the apex of the stem and terminate with first a bottleneck and then a rounded tip. Setae in the central zone are directed downwards, while the outer ones are directed towards the retrodorsal margin. They are closely spaced (∼25 μm) in the same row, while there is a distance of ∼50 μm between one line and the next.

**k3:** The shaft ends in a pointed apex with little expansion, giving the setae a "clock hand" appearance. The setae emerge from the cuticula and are oriented towards the retrodorsal margin. Adjacent setae on the same raw have a distance of about 15 μm, while between raws the distance is about double.

**k4:** The shaft is straight or curved. Lateral expansions emerge at about 1/5 of the stem length and are poorly developed, ending in a rounded tip. The setae move towards the retrodorsal margin of the tibia. The setae are close together. They are 20 μm apart on the same raw and 30 μm apart among raws.

**k5:** Just beyond the basal part of the shaft, flattened expansions emerge from its sides, running along its entire length and ending in a truncated tip. The setae are all oriented

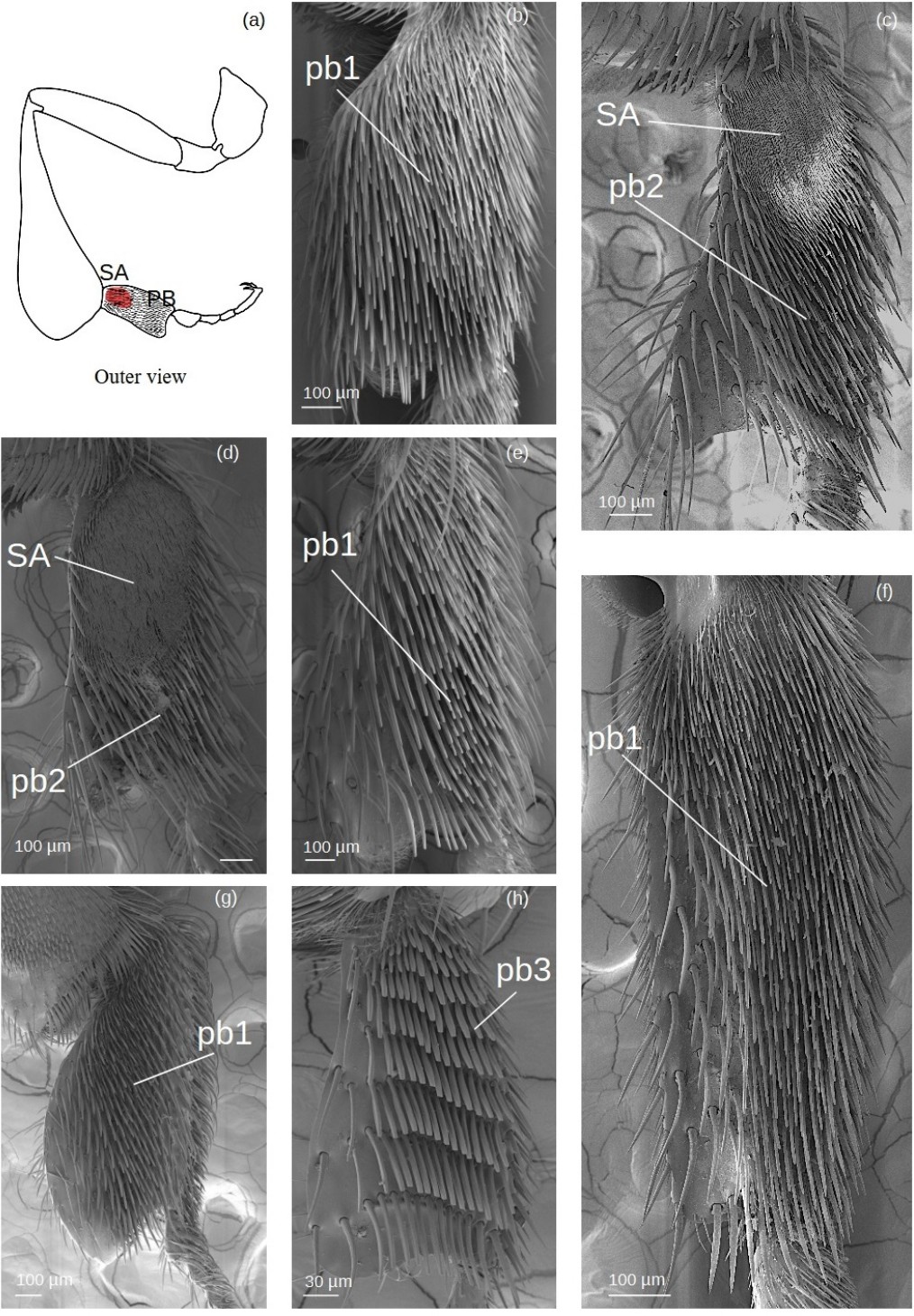

**Figure 7** **SEM microphotographs of the pollenbrush (PB) and sericeus area (SA).** (A) Diagram showing the location of the pollen brush and sericeus area in the hind basitarsus. (B) **pb1** type in *M.* cf. *eburnea*. (C) **pb2** type and sericeus area in *T.* cf. *hypogea* and (D) *T. dallatorreana*. (E) **pb1** type in *P. testacea*, (F) *Lestrimelitta* sp. and (G) *S.* cf. *latitarsis*. (H) Pollen brush type **pb3** of *T.* cf. *atomaria*.

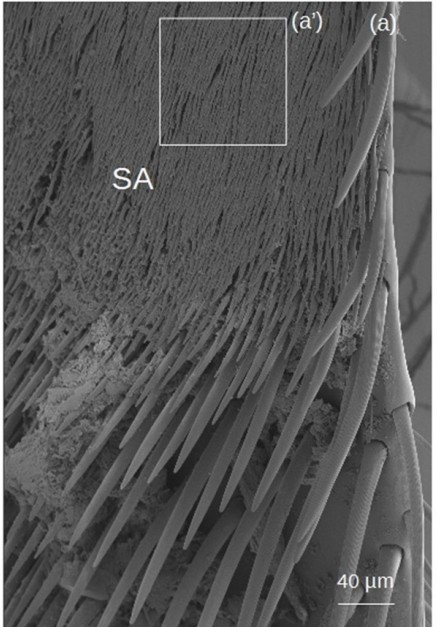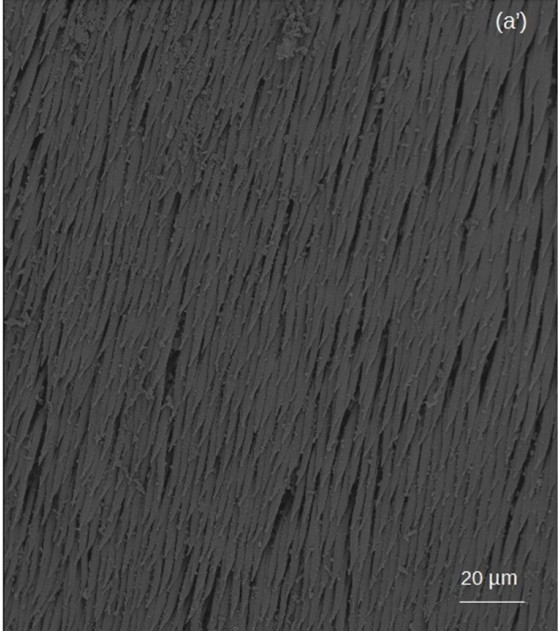

**Figure 8** **SEM microphotographs of the sericeus area (SA).** (A) Portion of the sericeus area of *T. dallatorreana*. (A') Enlargement of a part to show its setae.

towards the retrodorsal margin of the tibia. Setae spacing is approximately 12 μm along the row and 14 μm between rows.

**k6:** Sinuous shaft, wider base narrowing into a pointed tip and with parallel grooves along its entire length. Just beyond the basal part of the shaft, flattened expansions emerge from its sides, running the length of it and ending with a lancet shape. The setae emerge from the cuticular surface and are oriented towards the restrodorsal margin of the tibia. The adjacent setae are approximately 25 μm apart, while there is an approximate distance of 40 μm between the rows.

**k7:** Shaft with a wide base narrowing into a pointed tip; usually straight, but some setae in the marginal part of the keirotrichia zone have it curved. Flattened lateral expansions emerge from the base of the shaft and run the length of it to its tip where they converge, forming a bottleneck before ending in a truncated shape. The setae are oriented towards the retrodorsal margin of the tibia, while those in the distal part of the keirotrichiate zone emerge from the cuticle and head downwards. The setae on the row are about 35 μm apart, while the rows are about approximately 55 μm apart.

## Location of the setae in hind tibia and basitarsus

The location of each type of seta is indicated for the various zones of the hind tibia and basitarsus (Tables 2–3). Among the simple setae, **ns**, **ts** and **hs3** (Fig. 2) are the most widespread, with **ts** and **hs3** also forming the structures involved in pollen handling as penicillae (*i.e.,* inferior- and superior parapenicillum; Figs. 4–5) and those in the auricular

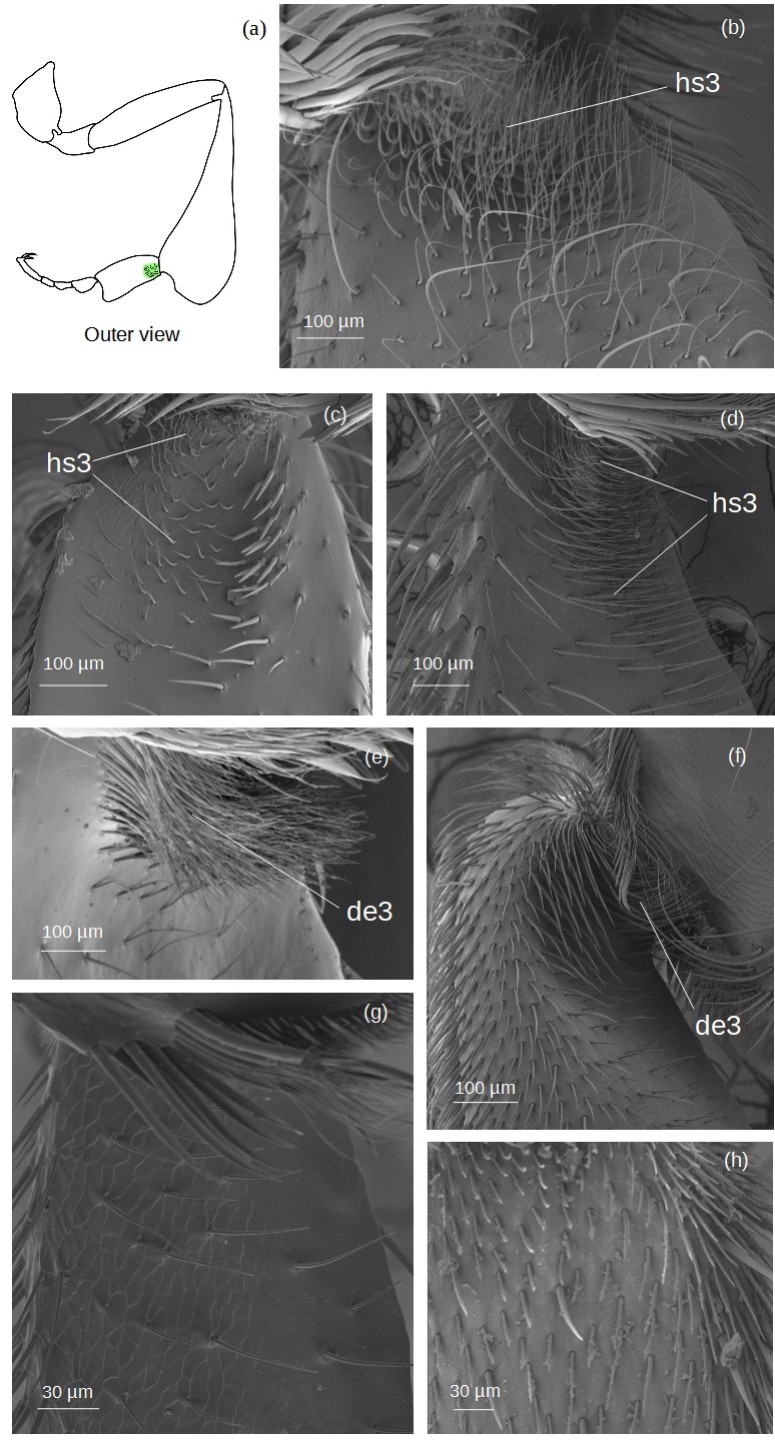

**Figure 9** **SEM microphotographs of the setae covering the auricular area.** (A) Diagram showing the location of the auricular area in the hind basitarsus. The auricular area is covered by **hs3** type setae in (B) *M.* cf. *eburnea,* (C) *T. dallatorreana* and (D) *P. testacea.* In (E) *T.* cf. *hypogea* and (F) *S.* cf. *latitarsis* **de3** type is observed. The auricular area is absent in (G) *T.* cf. *atomaria* and (H) *Lestrimelitta* sp.

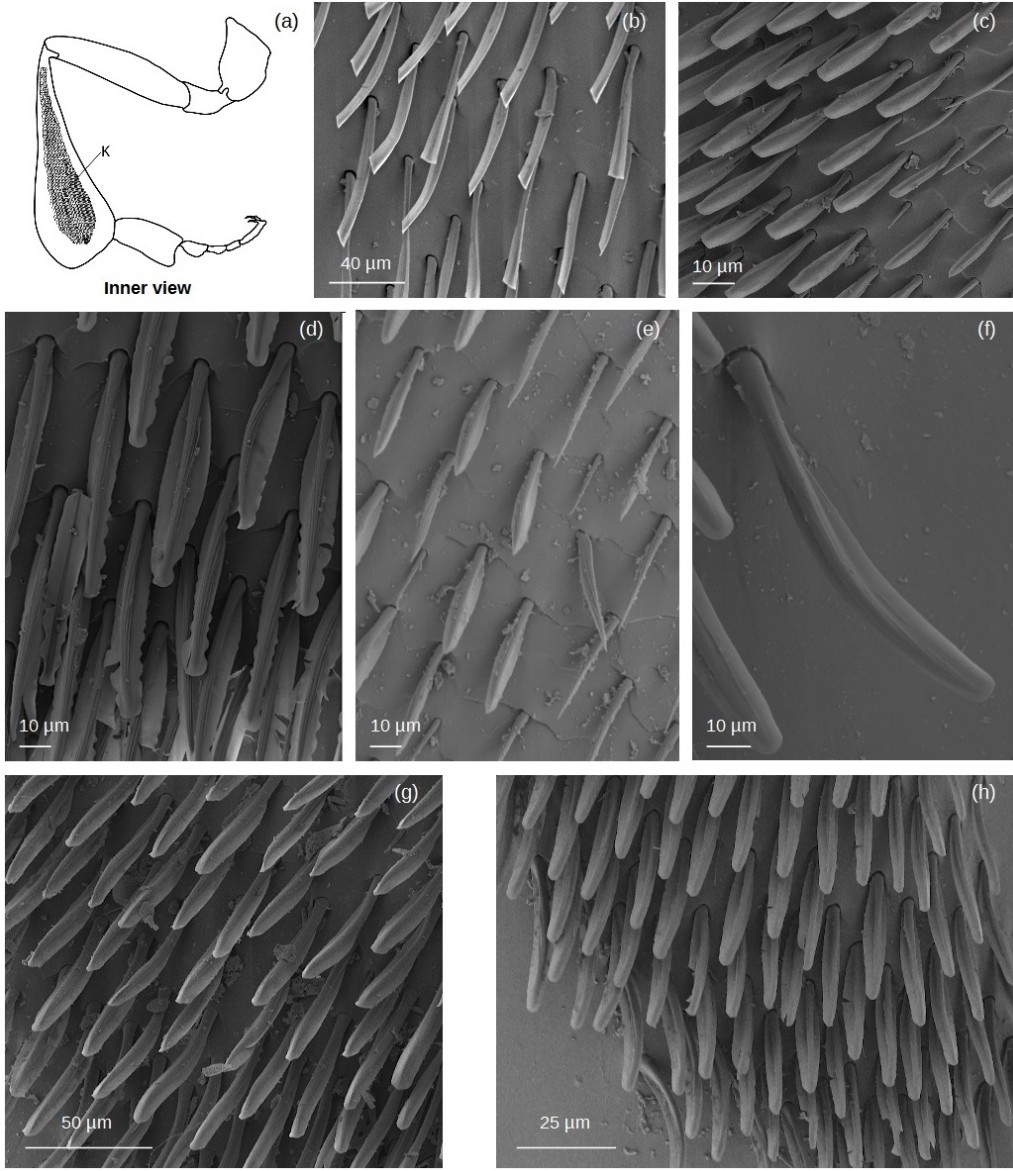

**Figure 10   SEM microphotographs of the keirotrichia (K).** (A) Diagram showing the location of the keirotrichiate zone on the retrolateral surface of the hind tibia. (B) **k1** type in *M.* cf. *eburnea*. (C) **k5** type in *T.* cf. *atomaria*. (D) **k2** type in *P. testacea*. (E) **k3** type in *S.* cf. *latitarsis*. (F) **k4** type in *T. dallatorreana*. (G) **k6** type in *Lestrimelitta* sp. (H) **k7** type in *T.* cf. *hypogea*.

area (Fig. 9). **Hs2** (Fig. 2) are exclusively located in the corbicula (Table 2), and **hs3** (Fig. 2) on the prolateral surface of the basitarsus (Table 3).

Regarding branched setae, the most common are the spinulate (**sp**) (Fig. 3; Tables 2–3). In contrast, plumose (**pl**) setae (Fig. 3) have a limited distribution on the tibia, being restricted to the retrodorsal fringe and distal margin, and are absent on the basitarsus (Tables 2–3). Dendritic setae (**de1–3**) (Fig. 3), on the other hand, exhibit a broader distribution on the basitarsus, with types **de1** and **de3** specifically covering the auricular

Table 2 Location of seta types in various zones of the hind tibiae of *Lestrimelitta* sp., *M.* cf. *eburnea*, *P. testacea*, *S.* cf. *latitarsis*, *T. dallatorreana*, *T.* cf. *hypogea* and *T.* cf. *atomaria*.

| Classification of setae | | HIND TIBIA | | | | | | | | | | | |
|---|---|---|---|---|---|---|---|---|---|---|---|---|---|
| | | Prolateral surface | | | | | Retrolateral surface | | | | Fringes | | |
| Type | Subtype | F | C | IP | P | SP | Sz | Az | K | R | PF | RF | DT |
| Simple setae | ns | + | + | | | | | + | + | | + | + | |
| | ts | + | | + | | | + | + | + | | + | + | |
| | hs1 | + | + | | | | | | | | | | |
| | hs2 | | + | | | | | | | | | | |
| | hs3 | | + | + | | + | + | + | | | + | + | |
| Spinulate | sp | + | | + | | | + | + | + | | + | + | |
| Pectinate | pe1 | + | | | | | | | | | | | |
| | pe2 | + | + | | | | | | | | | + | + |
| Plumose | pl | | | | | | | | | | | + | + |
| Dendritic | de1 | | | | | | | | | | | | + |
| | de2 | | | | | | | | | | | | |
| | de3 | | | | | + | | | | | | + | + |
| Keirotrichia | ke1 | | | | | | | | + | | | | |
| | ke2 | | | | | | | + | + | | | | |
| | ke3 | | | | | | | | + | | | | |
| | ke4 | | | | | | | + | + | | | | |
| | ke5 | | | | | | | | + | | | | |
| | ke6 | | | | | | | | + | | | | |
| | ke7 | | | | | | | + | + | | | | |
| Penicillum | pen1 | | | | + | | | | | | | | |
| | pen2 | | | | + | | | | | | | | |
| | pen3 | | | | + | | | | | | | | |
| Rastellum | ra1 | | | | | | | | | + | | | |
| | ra2 | | | | | | | | | + | | | |
| | ra3 | | | | | | | | | + | | | |
| | ra4 | | | | | | | | | + | | | |
| | ra5 | | | | | | | | | + | | | |

Notes.

F, Fundus; C, Corbicula; IP, Inferior parapenicillum; P, Penicillum; SP, Superior parapenicillum; Sz, Supraglabrate zone; Az, Apical glabrate zone; K, Keirotrichiate zone; R, Rastellum; PF, Proventral fringe; RF, Retrolateral fringe; DT, distal margin. +, Presence.

area (Table 3). The penicillum (Fig. 4), the rastellum (Fig. 6), the pollen brush (Fig. 7), the sericeus area (Figs. 7–8) and the keirotrichia (Fig. 10) consist of setae exclusive to these structures and are not found in other areas of the tibia and basitarsus (Tables 2–3). The exception is **ke5** (Fig. 10), which covers the keirotrichiate zone (Table 2) and forms the pollen brush (Fig. 7, Table 3).

## Distribution of setal types throughout species

Variations in the types and location of setae were observed among the species (Table 4, Fig. 11). The highest diversity of setae is exhibited by *T.* cf. *hypogea* and *T. dallatorreana* (Table 4). In contrast, *Lestrimelitta* sp. and *T.* cf. *atomaria* exhibit the lowest diversity,

Table 3   Location of seta types in the various zones of the hind basitarsi of Lestrimelitta sp., *M.* cf. *eburnea, P. testacea, S.* cf. *latitarsis, T. dallatorreana, T.* cf.*hypogea* and *T.* cf. *atomaria.*

| Classification of setae | | HIND BASITARSUS | | | | |
| --- | --- | --- | --- | --- | --- | --- |
| | | Prolateral surface | | Retrolateral surface | | Distal margin |
| Type | Subtype | AA | Ps | SA | Rs | DB |
| Simple setae | ns | | + | | | |
| | ts | | + | | + | |
| | hs1 | | + | | | |
| | hs2 | | | | | |
| | hs3 | + | + | | | |
| Spinulate | sp | | + | | + | |
| Pectinate | pe1 | | | | | |
| | pe2 | | | | + | |
| Plumose | pl | | | | | |
| Dendritic | de1 | + | | | + | + |
| | de2 | | + | | | + |
| | de3 | + | + | | | + |
| Keirotrichia | ke1 | | | | | |
| | ke2 | | | | | |
| | ke3 | | | | | |
| | ke4 | | | | | |
| | ke5 | | | | + | |
| | ke6 | | | | | |
| | ke7 | | | | | |
| Pollen brush | pb1 | | | | + | |
| | pb2 | | | | + | |
| | pb3 | | | | + | |
| Sericeus area | SA | | | + | | |

Notes.
AA, Auricular area; Ps, Prolateral surface; SA, Sericeus area; Rs, Retrolateral surface; DB, distal margin. +, Presence.

with only eight types of setae (Table 4). *M.* cf. *eburnea, P. testacea* and *S.* cf. *latitarsis* are intermediate cases, displaying ten, eleven and twelve types of setae, respectively (Table 4). The simple **ns** and **ts** (Fig. 2) setae are present in all species (Fig. 11), whereas **hs2** (Fig. 2) are observed exclusively in the corbicular cavity of *T.* cf. *hypogea* and *P. testacea.* (Fig. 11 and Fig. S2).

Regarding the setae in the hind tibia, the main differences between species are observed along the proventral and retrodorsal fringes, in the corbicula, the keirotrichiate zone and in the distal margin (Fig. 11).

Proventral and retrodorsal fringes. They may have a few simple setae (*e.g.*, **ts** or **hs3**) (Fig. 2), as showed in *T.* cf. *atomaria* and *S.* cf. *latitarsis*, or a combination of simple and branched setae (*e.g.*, **ts** or **sp**) (Figs. 2–3), as in *P. testacea* and *Lestrimelitta* sp. (Fig. 11). In *M.* cf. *eburnea*, both fringes consist of dense simple setae **hs3** (Fig. 2 and Fig. S2). In contrast, in *T. dallatorreana* and *T.* cf. *hypogea*, dense branched setae **pe2** (Fig. 3) mainly cover the retrodorsal margin (Fig. 11 and Fig. S2). Corbicula. It is almost entirely glabrous,

**Table 4  Distribution of setal types among stingless bees.**

| Classification of setae | | Stingless bee species | | | | | | |
|---|---|---|---|---|---|---|---|---|
| Type | Subtype | *M.* cf. *eburnea* | *P. testacea* | *S.* cf. *latitarsis* | *T. dallatorreana* | *T.* cf. *atomaria* | *T.* cf. *hypogea* | *Lestrimelitta* sp. |
| Simple setae | ns | + | + | + | + | + | + | + |
| | ts | + | + | + | + | + | + | + |
| | hs1 | + | + | + | + | | | |
| | hs2 | | + | | | | + | |
| | hs3 | + | + | + | + | + | + | |
| Spinulate | sp | | + | + | + | | + | + |
| Pectinate | pe1 | | | | + | | + | |
| | pe2 | | | + | + | | + | + |
| Plumose | pl | | | + | | | | + |
| Dendritic | de1 | | | + | + | + | + | + |
| | de2 | + | | | | + | | |
| | de3 | + | + | | | | + | |
| Keirotrichia | | k1 | k2 | k3 | k4 | k5 | k6 | k7 |
| Penicillum | | pen1 | pen2 | pen2 | pen3 | pen2 | pen3 | |
| Rastellum | | ra1 | ra2 | ra3 | ra1 | ra4 | ra5 | |
| Pollen brush | | pb1 | pb1 | pb1 | pb2 | pb3(k5) | pb2 | pb1 |
| Sericeus area | | | | | SA | | SA | |

**Notes.**

+, Presence.

with a few scattered simple setae **hs1** (Fig. 2) as in *M.* cf. *eburnea*, a pair of **hs2** (Fig. 2) exclusively in *P. testacea* and *T.* cf. *hypogea*, or **ns** and **ts** types (Fig. 2) as in *T.* cf. *atomaria* and *Lestrimelitta* sp. (Fig. 11). On the other hand, simple **hs1** setae are more numerous in the corbicula of *T. dallatorreana*, as well as **ts** and **hs3** types in the corbicula of *S.* cf. *latitarsis* (Fig. 11 and Fig. S2). *T.* cf. *hypogea* has the hairiest corbicula, with simple **hs2** and **hs3** (Fig. 2) and branched setae **pe2** (Figs. 3 and 11). Keirotrichiate zone. Each species has a different type (**k1-k7**; Fig. 10– 11), which may occupy a very large area of the retrolateral surface of the tibia, as in *M.* cf. *eburnea*, *P. testacea*, *S.* cf. *latitarsis* and *Lestrimelitta* sp., or a smaller area as in *T.* cf. *atomaria* (Fig. 11 and Fig. S3). In *T. dallatorreana* and *T.* cf. *hypogea* it covers a narrow overlying stripe (Fig. 11 and Fig. S3). Distal margin. It is hairy in *M.* cf. *eburnea*, *S.* cf. *latitarsis*, *T. dallatorreana* and *T.* cf. *hypogea*, being composed of dense dendritic setae of type **de1**-**de3**, **de1** and pectinate **pe2** (Fig. 3), respectively (Fig. 11 and Fig. S3).

With respect to the hind basitarsus, differences between stingless bee species are observed at the level of the auricular area (Fig. 9), the retrolateral surface and the distal margin (Fig. 11). Auricular area. It is covered with thick **hs3** and **de3** (Figs. 2–3) seta types in *M.* cf. *eburnea*, *P. testacea*, *S.* cf. *latitarsis*, *T. dallatorreana* and *T.* cf. *hypogea*, whereas the tuft of setae is absent in *T.* cf. *atomaria* and *Lestrimelitta* sp. (Figs. 9, 11 and Fig. S4). Retrolateral surface. In many species it is completely covered by the pollen brush (Figs. 7, 11 and Fig. S5), except in *T. dallatorreana* and *T.* cf. *hypogea*, where it is reduced by the

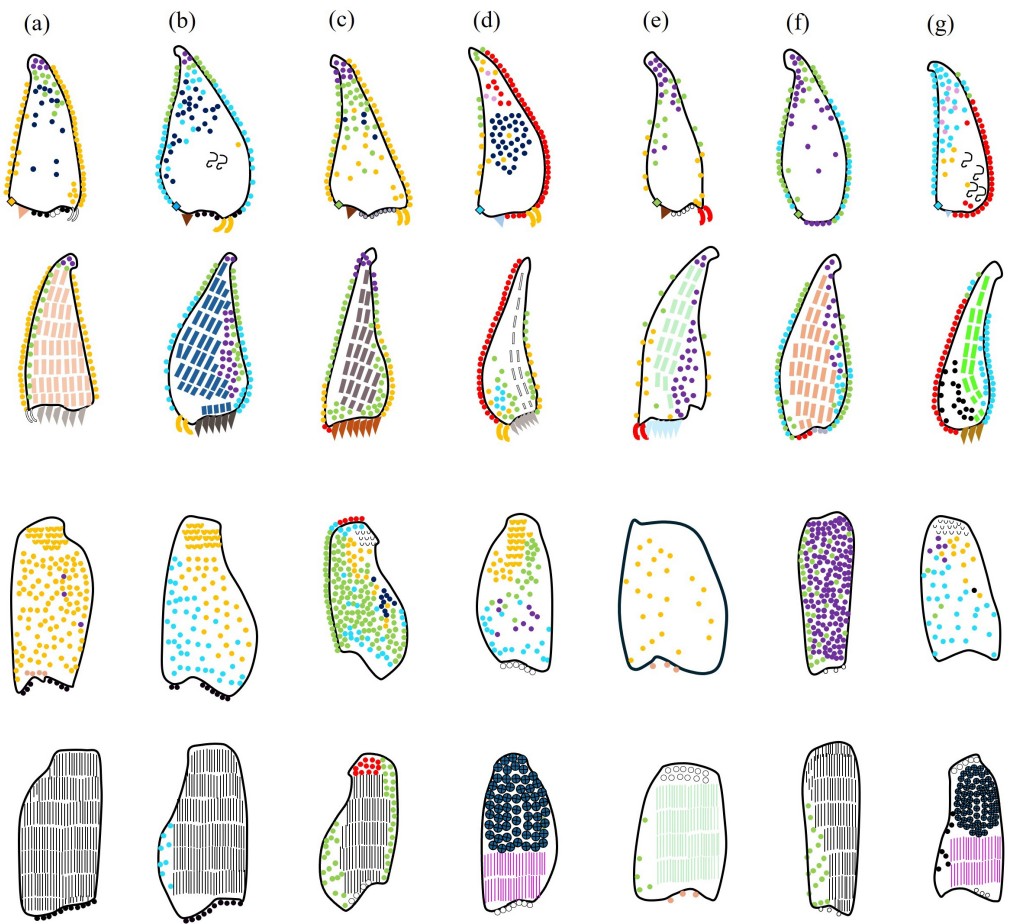

Simple setae: ● ns; ● ts; ● hs1; ϱ hs2; ● hs3. Branched setae: ● sp; ● pe1; ● pe2; ● pl; ○ de1; ● de2; ● de3.
Keirotrichia: ┃k1; ┃k2; ┃k3; ┃k4; ┃k5; ┃k6; ┃k7. Inf. parapenicillum: ◇.
Penicillum: ▼pen1; ▼pen2; ▼pen3. Sup. parapenicillum: )). Rastellum: ◖ra1; ◖ra2; ◖ra3; ◖ra4; ◖ra5.
Pollen brush: ▓ pb1; ▓ pb2; ▓ pb3. Auricular area: ▰. Sericeus area: ●.

**Figure 11 Drawing of the distribution of setal types on the hind tibiae and basitarsi of the stingless bees.** Tibia (outer and inner views) and basitarsus (outer and inner views) of each species, from the top to the bottom. (A) *M.* cf. *eburnea* (B) *P. testacea* (C) *S.* cf. *latitarsis* (D) *T. dallatorreana*; (E) *T.* cf. *atomaria* (F) *Lestrimelitta* sp. (G) *T.* cf. *hypogea*.

presence of the sericeous area (Figs. 7–8). In *T.* cf. *atomaria*, the setae of the pollen brush (**k5** type) are arranged in distinct parallel rows (Fig. 7). Distal margin. Here, there may be a few dendritic setae (*e.g.*, **de1** and **de2**) (Fig. 3) as in *T.* cf. *atomaria*, *Lestrimelitta* sp. and *T.* cf. *hypogea* (Fig. 11 and Fig. S5), or numerous (*e.g.*, **de1** and **de3**) (Fig. 3) as in *M.* cf. *eburnea*, *P. testacea*, *S.* cf. *latitarsis* and *T. dallatorreana* (Fig. 11 and Fig. S5).

Table 5  Overview of setal types and degree of development of penicillae and rastella across stingless bees.

| | *M. cf. eburnea* | *P. testacea* | *S. cf. latitarsis* | *T. dallatorreana* | *T. cf. atomaria* | *T. cf. hypogea* | *Lestrimelitta sp.* |
|---|---|---|---|---|---|---|---|
| **Inferior parapenicillum** | **hs3** | **sp** | ts | sp | ts | sp | ts |
| **Penicillum** | **pen1** | **pen2** | **pen2** | **pen3** | **pen2** | pen3 | * |
| **Superior parapenicillum** | de1 | hs3 | **hs3** | hs3 | **pe2** | * | * |
| **Rastellum** | **ra1** | **ra2** | **ra3** | **ra1** | ra4 | ra5 | * |
| **Auricular area** | **hs3** | **hs3** | **de1** | hs3 | * | de1 | * |

Notes.
 Bold text indicates well-developed structures. Normal text indicates reduced structures.
 *Absence of the structure.

## Structural variability in penicillae, rastella and auricular areas among species

The types of setae and the degree of development of the penicillae, rastella and auricular areas are highly variable among species (Table 5). The lengths of setae forming these structures are listed in Table S5.

## DISCUSSION

In stingless bees, the morphological diversity of the setae and structures involved in pollen handling of the hind tibiae and basitarsi are poorly investigated, especially through a comparative approach. This finding is unexpected given the wide diversity of foraging behaviors within the group, including floral pollen collection, kleptoparasitism and necrophagy, which provides a framework to investigate how these structures vary among species that collect floral pollen and those that have secondarily abandoned this behavior. This work provides for the first time a detailed SEM study of these setae in the floral pollen collector *M.* cf. *eburnea*, *P. testacea*, *S.* cf. *latitarsis*, *T. dallatorreana*, *T.* cf. *atomaria*, in the kleptoparasite *Lestrimelitta* sp. and in the obligate necrophagous *T.* cf. *hypogea* stingless bees, from Peru.

First of all, we showed that three types of simple setae—**ns**, **ts**, and **hs3** (Fig. 2)—are highly conserved among the species (Table 4), as are their locations along the tibia (*i.e.*, fundus, proventral and retrodorsal fringes, close to the keirotrichiate zone) and basitarsus (*i.e.,* prolateral and retrolateral surfaces) (Tables 2–3, Fig. 11). This aligns with *Wille (1979)* recognition of simple setae along the retrodorsal fringe of the tibia as a primitive feature in Meliponini. Indeed, this condition is observed in various fossil specimens, including *Cretotrigona prisca* Michener & Grimaldi 1988 (New Jersey amber, Cretaceous, approx. 65–96 Mya); *Exebotrigona velteni* Engel & Michener 2013; *Kelneriapis eoceanica* Kelner-Pillault 1969; *Liotrigonopsis rozeni* Engel 2001 (Baltic amber, Eocene, approx. 34–55 Mya); *Austroplebeia* genus Moure (Chinese amber, Miocene, approx. 14 Mya); *Proplebeia dominicana* Wille & Chandler 1964; *Proplebeia tantilla* Camargo et al. 2000; *Proplebeia vetusta* Camargo et al. 2000 (Dominican Republic amber, Miocene, approx. 15–20 Mya); and *Axestotrigona kitingae* Engel & Solórzano-Kraemer 2022 (Tanzanian

copal, Holocene, approx. 987–1051 Kya). Coherently, these simple setae have also been reported in other genera of Meliponini, such as on the retrodorsal and proventral fringes of the tibia of *Friesella schrottkyii* Friese 1900 (*Alves, Patricio & Soares, 2003*) and in the African stingless bees *Geniotrigona lacteifasciata* Cameron 1902, *Tetragonula melanocephala* Gribodo 1893 and *Tetragonula sirindhornae* Michener & Boongird 2004 (*Zubaidah et al., 2017*). Therefore, these setal types are plesiomorphic traits, which do not exhibit structural modifications associated with specific foraging strategies. The morphology of these simple setae suggests that they could represent trichoid sensilla acting as mechanoreceptors (*Queiroz Fialho et al., 2014*).

A potential functional adaptation is likely represented by the **hs3** setae (Fig. 2), which form a dense fringe along the margins of the tibia, bending towards the corbicular concavity, as seen in *M*. cf. *eburnea*, or curving to close the corbicula at the distal margin of the tibia, as in *S. latitarsis* (Fig. S2). This suggests that specialized pollen transport in floral pollen collector species has evolved from a primitive state, characterized by a few simple, outward-facing setae, as observed in *T*. cf. *atomaria*, to a state with long, curved setae directed towards the corbicula, as in *S*. cf. *latitarsis,* and the development of very dense fringes along both tibial margins, as evidenced in *M*. cf. *eburnea* (Fig. S2). This structural specialisation may have been driven by the mechanical advantages offered by the long, curved setae, which help to fix and support the pollen load within the corbicular cavity.

Other simple setae, *i.e.,* types **hs2** (Fig. 2), are observed exclusively in the corbicular cavity of *T*. cf. *hypogea* and *P. testacea* (Fig. 11 and Fig. S2). While their function is still unclear, their positioning suggests a sensory role like the single spindle hairs in *A. mellifera*, which help determine the maximum pollen pellet volume (*Hodges, 1967*). In *T*. cf. *hypogea* this may be a trait retained from the species' necrophagous behavior, potentially useful for transporting other materials within the corbicula.

In contrast to the simple setae, most types of branched setae exhibit considerable interspecific structural variations (Fig. 11), likely reflecting specialization in pollen handling. We observed that spinulate **sp** setae and the dendritic **de1** type (Fig. 3) are the most common branched setae (Table 4), exhibiting a conserved positional pattern across the species (*i.e.,* along both margins of the tibia and on the prolateral surface of the basitarsus, respectively) (Fig. 11). *Alves, Patricio & Soares (2003)* reported the same type of setae on the proventral and retrodorsal fringes of the tibia in *P. cupira*.

According to *Thorp (1979)*, branching increases the surface area of setae and facilitates pollen capture in many bee species. Furthermore, a higher density of branched setae allows specialisation on certain pollen types (*Thorp, 1979*; *Portman & Tepedino, 2017*). Therefore, the **sp** and **pe2** setae (Fig. 3), located along the margins of the corbicula in *P. testacea* and *T. dallatorreana*, respectively (Fig. S2) could efficiently retain pollen compared to the simple setae found in other floral pollen collectors. The spiniform branching of **sp** setae may enhance pollen capture through an impalement mechanism, anchoring individual grains.

Setal branching diversification in floral pollen collector species is most evident along the distal margins of the tibia and basitarsus (Fig. 11). In these regions, we primarily observed dendritic setae (**de1-de3**) (Fig. 3), whose consistent presence suggests a specialized function,

likely related to efficient pollen capture. Specifically, the pollen may be trapped on the distal parts of the tibia, which come into contact with the corbicula, and on the basitarsus, which closes the pollen brush. This results in a mechanical action that facilitates the retention of the corbicular load and the pollen in the brush. Therefore, the presence of setae with longer and more flexible branches on the distal tibial margin of *T. dallatorreana* (**pe2**), *S.* cf. *latitarsis* (**pl**), *M.* cf. *eburnea*, and *P. testacea* (**de3**) (Figs. 3 and 11) may enhance pollen retention in their corbiculae. In *M.* cf. *eburnea* and *P. testacea*, this feature is further complemented by a covering of **de3** setae on the distal margin of the basitarsus (Fig. S4), suggesting greater overall efficiency in retaining pollen within the pollen brush.

Among the floral pollen collector species, *T.* cf. *atomaria* appears to be the least specialized for trapping pollen within the pollen brush, as indicated by the sparse presence of **de2** setae (Fig. 3) in the distal margin of the basitarsus (Fig. S4). It is noteworthy that the dendritic setae are the only branched setae types observed in this species (Table 4). Given the basal phylogenetic position of this genus (*Rasmussen & Camargo, 2008*), it remains to be determined whether these setae represent an ancestral condition from which other branched setae types evolved.

The presence of a dense cover of **pe2** and **de1** setae (Fig. 3) on the retrodorsal surface of the basitarsus, near the tibia–basitarsus junction, in *S.* cf. *latitarsis* and *T.* cf. *atomaria* respectively (Fig. 11 and Fig. S5), appears to represent specialization in these species. Whether these setae play a role in assisting corbicula loading during tibia–tarsus joint movement remains to be clarified.

Finally, some branched setae are shared with other corbiculate bees, such as *Bombus impatiens* Cresson 1863, where **sp** setae surround the corbicula and plumose setae (pl) are found at the tibia-tarsal junction (*Hines et al., 2022*). Further investigation may clarify whether these are indeed the same setae and whether they are shared or independently evolved traits.

From the perspective of branched setae, we generally do not observe marked divergences among the floral pollen collectors, the cleptoparasite and necrophagous bees (Fig. 11). In fact, *T.* cf. *hypogea* retains a branched setae arrangement that closely resembles that of its congener *T. dallatorreana* (Table 4, Fig. 11). Moreover, *T.* cf. *hypogea* exhibits the potential specialization at the tibia–tarsus joint as showed for *T.* cf. *atomaria*, because the presence of **de1** setae (Fig. 11). In contrast, *Lestrimelitta* sp. appears to have converged on a tibial setal pattern similar to that of the phylogenetically distant *P. testacea* (*Rasmussen & Camargo, 2008*) (Table 4, Fig. 11). Nevertheless, in *Lestrimelitta* sp. and *T.* cf. *hypogea*, the distal margin of the basitarsus is nearly devoid of setae—a condition, also observed in *S.* cf. *latitarsis* and *T.* cf. *atomaria* (Fig. 11)—which may indicate a less specialized state, considering the previously discussed function to retain pollen in the pollen brush.

The structures involved in pollen handling and corbicular loading, such as the **penicillae** (Figs. 4–5), the **rastellum** (Fig. 6), the **pollen brush** (Fig. 7) and the setae of the **auricular area** (Fig. 9), show distinct evolutionary trends, as some are conserved across species, while others show greater variability, reflecting a probable specialization (Table 4). All floral pollen collector species possess a well-developed **penicillum** (Table 5). The **pen2** type (Fig. 4) is the most widespread (Table 4), whereas the **pen3** of *T. dallatorreana* (Fig. 4)

appears highly specialized: its curved and branched setae are presumably more effective both in directing pollen upward during tibio-basitarsal movements and in trapping pollen grains. Also, both the **inferior**- and **superior parapenicillum** (Figs. 4–5) are present in all floral pollen collector species, albeit with varying degrees of development (Table 5). It is noteworthy that the branched setae of the inferior parapenicillum in *P. testacea* and the superior parapenicillum in *T.* cf. *atomaria* (Figs. 4–5) may enhance pollen capture, thereby complementing the action of the penicillum.

The **rastellum** is the structure with the greatest morphological variability among species, likely indicating a trait that has specialized in different adaptive pathways. This variability is observed both in the type of setae and in its structural development (Table 5). While many species (*i.e., M.* cf. *eburnea, P. testacea* and *S.* cf. *latitarsis*) possess a well-developed rastellum, intermediate forms (*i.e., T. dallatorreana* and *T.* cf. *hypogea*) and a significantly reduced rastellum (in *T.* cf. *atomaria* ) can be found (Fig. 6). In addition, there are variants with branched setae, as in *P. testacea* (type **ra2**) and *S.* cf. *latitarsis* (type **ra3**) (Fig. 6), which may indicate a derivation from the single setae rastella. These adaptations have probably improved the efficiency of pollen collection and trapping.

In contrast, the **pollen brush** is morphologically conserved among floral pollen collector species (Table 4), typically conforming to the **pb1** type (Fig. 7). In most species, it extends over the entire retrodorsal surface of the basitarsus (Fig. S5). An exception is *T. dallatorreana*, in which the brush is reduced due to the presence of a sericeous area (Fig. 7). The pollen brush of *T.* cf. *atomaria* is characterized by a lower density of setae arranged in distinct parallel rows (Fig. 7), in contrast to the densely packed setae of *M.* cf. *eburnea* and *P. Testacea* (Fig. 7), suggesting that the denser setae may confer greater efficiency in pollen capture during the brushing process.

Setae in the **auricular area** show limited variation (Fig. 11), with three main types identified (Fig. 9). The **hs3** type (Fig. 2) is the most frequent, occurring in *M.* cf. *eburnea, P. testacea*, and *T. dallatorreana* (Fig. 9). An exception is *S.* cf. *latitarsis*, which displays a dense cover of **de1** setae (Fig. 3) in this region (Fig. 9). These setae likely represent a specialized condition and may participate in collecting pollen deposited on the leaf or the ground, as proposed by *Roubik (2018)*.

Complementing these morphological patterns, this study provides the first SEM-based description of the setae forming the **sericeus area** (Figs. 7–8), a trait considered diagnostic of the genera *Trigona* and *Tetragonisca* (*Michener, 2007*). Although its precise role in pollen handling remains unclear, its distinctiveness and apparent association with modified pollen brushes suggest a functional relevance that warrants further investigation.

The cleptoparasite *Lestrimelitta* sp. and the necrophagous *T.* cf. *hypogea* exhibit contrasting morphological features, particularly in the setal types (Fig. 11) and in the presence and degree of development of structures associated with pollen handling (Table 5).

*Lestrimelitta* sp. displays a markedly simplified tibial and basitarsal morphologies (Figs. S2 and S4), characterized by a flattened corbicula, reduced setal diversity (Table 4), and the absence of key structures involved in pollen handling, such as the penicillum, the rastellum and the setae of the auricular area (Table 5). These reductions may suggest a relaxation of selective pressures associated with floral pollen harvesting. Although direct

evidence is lacking as to whether the corbicula is still used for transporting stolen pollen—as reported for *Lestrimelitta limao* (*Zuben & Nunes, 2014*)—it is conceivable that this structure remains functional in transport, possibly through simplified **Type I ipsilateral** movements that do not require the coordination of penicillum and rastellum. By contrast, *T.* cf. *hypogea* retains a more complex tibio-basitarsal morphology (Figs. S2 and S4), including a higher diversity of setae- simple and branched (Table 4)- and partial conservation of pollen-related structures (Table 5). In fact, while the superior parapenicillum is absent, both the penicillum and rastellum are present, albeit in reduced form (Figs. 4 and 6, Table 5). In addition, this species displays dense setal coverage in the auricular area (Fig. 9) and retains the sericeus area (Fig. 8), the diagnostic character of the genus (*Michener, 2007*). This anatomical configuration suggests that *T.* cf. *hypogea* may still be capable of performing both **Type I** and **Type II** movements, enabling a broader repertoire of manipulative behaviors compared to *Lestrimelitta* sp. The persistence of these traits despite the apparent shift away from pollen collection probably reflects the maintenance of adaptations that may still be useful for transporting resin and other nest-building materials.

Finally, we categorize the types of **keirotrichia** (Fig. 10), which, although not involved in pollen handling, remain poorly understood in *Meliponini*. These setae exhibit significant variability across species (Table 4). In *Trigona* species, the keirotrichia covers a narrow area raised from the tibial surface, forming a supraglabrate zone along the retrodorsal margin, whereas in all other species it is very extensive over the entire retrolateral surface of the tibia (Fig. S3). Future research could explore how microtrichia types - setae located on bee wings - wing vein patterns, and wing morphology influence keirotrichia architecture, potentially shedding light on the evolutionary pressures shaping grooming adaptations.

The existing literature does not provide clarity on the link between a ''reduced corbicula'' and a ''functional corbicula''. A reduced corbicula is reported for *T. hypogea* by *Roubik (1982a)*, *Roubik (1989)* and *Camargo & Roubik (1991)*. For the genus *Lestrimelitta*, numerous authors have reported its non-functionality (*Kerr, 1951*; *Sakagami & Laroca, 1963*; *Wille, 1983*; *Michener, 2007*). However, *Zuben & Nunes (2014)* provided evidence of *Lestrimelitta limao* using its corbicula to load pollen under laboratory conditions. In the light of what has been discussed above, we propose a reconsideration of the definition of the 'functional corbicula,' as a reduction or flattening of the surface alone is insufficient to conclude that it is no longer used for transporting pollen or other materials. Even in the absence of penicillum, setae in the auricular area, and rastellum, stingless bees could still potentially utilize it through Type I movements, which do not depend on these structures. Furthermore, the corbicula is typically defined as a glabrous, concave area bordered by long fringes of hairs, usually used for pollen transport (*Michener, 1999*). However, this definition fails to fully capture the complexity of this multifunctional structure, particularly in Meliponini. A critical aspect often overlooked is the diversity and functionality of the tibial and basitarsal setae, which should not be regarded as accessory components but rather as integral parts of a unified structure. Therefore, corbicula is not a static concavity, but a dynamic system that integrates sensory feedback to optimize its functionality. In both the ancestral condition (*T.* cf. *atomaria*) and in cases of secondary reduction (*Lestrimelitta* sp.), the corbicula appears reduced and nearly or fully flattened,

with low setal diversity, potentially maintaining its ability to transport pollen using simple Type I movements. In contrast, other species (*M.* cf. *eburnea*, *P. testacea*, *S.* cf. *latitarsis* and *T. dallatorreana*) exhibit a greater diversity of setae and more developed structures enabling more complex Type II movements. These considerations highlight the importance of a functional perspective when examining the corbicula, and caution against overinterpreting its reduction or absence in purely morphological terms.

This work provides the first comprehensive analysis of setae and structures involved in pollen handling on the hind tibia and basitarsus of stingless bees, providing a reference for future research on morphology, functional ecology in Meliponini, and setal diversity in other corbiculate bees. Although the sample includes only a few specimens, our results highlight the need for further investigation into the functional role of these setae and the phylogenetic and environmental drivers that have shaped their differentiation. It is challenging to determine how the hind tibia and basitarsus, together with their setae, have evolved in different lineages, as various selective pressures may have been at work, including those not directly related to pollen collection. Studying how the hind tibia and basitarsus, together with their setae, have evolved in different lineages is no easy task, as various selective pressures may have been at work, including those not directly related to pollen collection. This is particularly relevant given that the tibia is a highly versatile structure that, thanks to keirotrichia, is also used for wing cleaning.

Future research should extend this analysis to other stingless bee species, focusing on the mechanical and sensory roles of different types of setae, while also investigating the evolutionary constraints and selective pressures that have shaped their development across Meliponini lineages.

## CONCLUSIONS

This study documents the morphological diversity of setae and structures involved in pollen handling in hind tibiae and basitarsi of Peruvian stingless bees. Through comparative SEM analysis between species of different lineages and foraging behaviour, we show that some setae are highly conserved, while others show great variability, possibly suggesting adaptive significance. We believe that the corbicula, as a versatile structure, is very complex and that its function must include clarification of the role of these setae. Further research is needed to extend the analysis to other Meliponini species and to elucidate the sensory role of these setae, how these traits evolve in different lineages and how they are used to implement pollen collection movements.

## ACKNOWLEDGEMENTS

We thank the staff of Urku Estudios Amazónicos and the students at the National University of San Martín for their participation in the sampling activities and their support with the planning and coordination of the fieldwork.

### Funding
This work was supported by the Department of Excellence of the University of Roma Tre, the Programa Nacional de Investigación Científica y Estudios Avanzados Proyecto "Modelo transdisciplinar para la comprensión de la diversidad clave de las abejas peruanas sin aguijón (Hymenoptera: Apidae: Meliponini) con fines de conservación y el desarrollo de una meliponicultura competitiva en la Amazonia" (PROCIENCIA no 82163 - PERU), the Cooperation Project of Sapienza 2023 (Prot. no 0001071) "Progettare un futuro per la meliponicoltura e le api native del Perù" and the NGO Estudios Amazónicos. The funders had no role in study design, data collection and analysis, decision to publish, or preparation of the manuscript.

### Grant Disclosures
The following grant information was disclosed by the authors:
Department of Excellence of the University of Roma Tre: PROCIENCIA no 82163 - PERU.
Cooperation Project of Sapienza 2023: Prot. no 0001071.
The NGO Estudios Amazónicos.

### Competing Interests
The authors declare there are no competing interests.

### Author Contributions
- Marilena Marconi conceived and designed the experiments, performed the experiments, analyzed the data, prepared figures and/or tables, authored or reviewed drafts of the article, and approved the final draft.
- Carlos Daniel Vecco-Giove analyzed the data, authored or reviewed drafts of the article, and approved the final draft.
- Javier Ormeño Luna analyzed the data, authored or reviewed drafts of the article, and approved the final draft.
- Agustín Cerna Mendoza analyzed the data, authored or reviewed drafts of the article, and approved the final draft.
- Emiliano Mancini analyzed the data, authored or reviewed drafts of the article, and approved the final draft.
- Andrea Di Giulio conceived and designed the experiments, performed the experiments, analyzed the data, authored or reviewed drafts of the article, and approved the final draft.

### Field Study Permissions
The following information was supplied relating to field study approvals (i.e., approving body and any reference numbers):

Field bee collections were permitted by the Servicio Nacional Forestal y de Fauna Silvestre del Peru (SERFOR) with official permits.

## Data Availability

The raw measurements are available in the Supplemental File.

## Supplemental Information

Supplemental information for this article can be found online at http://dx.doi.org/10.7717/peerj.19749#supplemental-information.

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
