# Peer review of "Scanning electron microscopy (SEM) reveals high diversity of setae on the hind tibiae and basitarsi of Peruvian Stingless Bees (Apidae: Meliponini)"

_PeerJ, doi:10.7717/peerj.19749_

## Round 0.1 · original submission · Major Revisions

The manuscript need careful revision prior to the resubmission.

Reviewer 1 ·

Basic reporting

This study involves extensive research defining setal types in legs of seven stingless bees, thus involves extensive descriptive data. Prior work performed similar analysis on 4 other stingless bee species. The greatest value of the work is the numerous SEMs showing setal morphology of these structures and recognizing the considerable diversity of seta that likely play a fairly complicated function in the food collection strategies of these bees. This could also inform the importance of setal morphologies more broadly across corbiculate bees and other bees in pollen collection.
The study, however, lacks a quantitative analysis of the data, providing thoughts on trends the authors note from descriptive data. This makes it difficult to take home key points. See Validity of Findings for more details on this.
Overall, I think this paper needs work on writing it in a way that is more focused on bigger picture take-home messages.
1) A lot of cross-species comparisons of morphology are described in detail in the discussion but could be moved instead to the results and instead the discussion could focus on discussing those comparisons and their implications and novelty more broadly.
2) The introduction could use more information about what the function of different setal types might be
3) The introduction should mention what the prior studies on stingless bee leg setal morphology did differently from the present study. (Line 95)
4) The study needs more of a hypothetical framework and more testing of hypotheses with analyses of the data.

Experimental design

The methods are mostly SEM imaging and setal descriptions. While this paper has descriptive value alone, I think it would be hard for anyone but the authors to use these fine distinctions on more specimens, as in some places the distinguishing features were not that clear.
Some specific issues with the design:
1) There is an assumption that one specimens represents all of a species. More clarity on sample sizes and this assumption is needed.
2) Line 128 – “main stem” needs to be defined. What is “approximate value” of branch lengths. Is this relative, absolute, and what is the value unit? Also, arguably, setal absolute length will be dependent on body size. Is there considerable body size variation in the tested bees or are they similar?
3) For some of the setal descriptions, it is hard to tell the difference between the categories. How is threadlike different from long and slender setae? What is “hair-like”? What is a “hair shape”? When I look at Figure 2, several of these do not look different to me.
4) Bristle vs. seta – may want to reflect on this distinction as some of the setal types look more likely classic bristles.
5) Ls2 – looks like it just has one length..what is the range?
6) Consider whether table could be used to show what is actually different between the simple setal types as it took awhile to see what is the essential contrast in this from the descriptions.
7) Line 256 – what is a “thorn” and how does that differ from a spiny process? What is a spicule vs. a spine.
8) From the description, what dendritic means different from other things is not clear to me.

Validity of the findings

In some cases there needs to be more analysis and more shown to aid in this interpretation. Some areas of concern here include:
1) One key stated goal of the paper is to note phylogenetic trends and conservation in setal morphology. However, there is no phylogeny or trait evolution inferred to make some of the inferences made in the conclusions.
2) Trait conservation could best be assessed using an analysis.
3) It is mentioned which are the main setal types found throughout all stingless bees for certain structures. It would be helpful if we could see a summary figure of the key types (maybe showing their morphology in the figure as it is difficult to follow the different setal types).
4) The other key take home is whether stingless bees that are cleptoparasitic or necrophagous have reduced or enhanced traits. This message would be more clear if the data were diagrammed (preferable phylogenetically) with an indication of which bees have each life history trait, showing the reduction and expansion in a figure.
5) There are tables with presence and absence of setal types in a given structure across Meliponini and of setal types overall by species. The types in a given structure by species is how data, in contrast, is best used to answer some of the questions above.
6) It could also be useful to do some kind of clustering analysis (PCA or phylogenetic) of setal type similarity by recording presence and absence of different types of setae by structure for different species and seeing which taxa (with different life history traits) have more similarity (e.g., is the cleptoparasite most deviant and is this unexpected considering phylogeny?).
7) The data could benefit from less fine dissection of differences (maybe greater clustering of setal types), as some of the seta look really similar to each other in morphology. For example, simple vs. branched could be analyzed.
8) Why not integrate data from the prior paper on 4 other species with this one to improve understanding of trends?

Additional comments

1) Line 65 – Setae need not be chemo or mechanosensory. Bristles often are and setae often are not innervated. None of the functions in the next line imply chemo or mechanosensation.
2) Line 26 - Abstract: necrophagus = necrophagy
3) Line 69 – Replace “basically” with “generally” or “In general”.
4) Line 94 - Unique to corbiculate bees or unique to Meliponini relative to other corbiculate bees? “in the group” is confusing here.
5) Line 113 – “In this work” sounds like a reference to the present study. Reword.
6) Table 1 – This is not really in table format.
7) Line 162 – Apis needs to be italicized.
8) Line 250 – “in contrast to” would be better than “whereas”
9) Line 389 – what does “exclusive in the corbicula” mean?
10) Line 486 – comma needed after latitarsis. This sentence was hard to read so could use some work.
11) Line 495 – Branching setae clearly are not derived in Meliponini as they are ancestral to bees. I think just one structure is meant to be referred to here?
12) Line 579 – I am not understanding the reference here to wing veins and shape - is this referring to the wings of the structure?
13) Line 646 – The reference to wings comes out of nowhere. Make the mention of alternative functions more clear early on.
14) Line 584 – Singular form is used here. Adjust to make the structure plural or add a “the” before the structures. This pertains to references of the rastellum and penicillum in later sentences.
15) Line 616 – 620 – I am not sure what is being referred to here. Are there certain lineages being referenced and what is ancestral vs. being reduced?

·

Basic reporting

The manuscript is an interesting paper due to its novel details and clear SEM figures. However, there are a few points that could be improved before publication, as outlined in the attachment.

Experimental design

no comment

Validity of the findings

The large amount of information on setae types and abbreviations makes them difficult to understand. Some types of setae exhibit similar characteristics and overlap in the range of lengths, so the definitions of the setae need to be clarified. Additionally, the methods for measurements should be made clearer.

Reviewer 3 ·

Basic reporting

This study sheds light on one of the most important topics in the taxonomy, ecology, and natural history of native stingless bees from the Neotropical region. The research presents robust data that characterize and describe numerous morphological structures essential for resource collection, highlighting their implications for ecosystem services such as pollination.

Experimental design

The study employed sophisticated methodologies, including high-resolution image capture through Scanning Electron Microscopy (SEM). Furthermore, these methods were applied to a considerable number of stingless bee species. The methodology was deemed appropriate to address the research questions.

Validity of the findings

The findings expand knowledge about structures never before characterized in various native bee species from the Neotropical region. They provide insights into the functions of these structures and propose significant implications for both the species’ life histories and applied fields such as meliponiculture.

Additional comments

The current version of the manuscript is well-structured and written. However, the suggestions provided throughout the text should be addressed to ensure the final version is publication-ready.

Annotated reviews are not available for download in order to protect the identity of reviewers who chose to remain anonymous.

---

## Round 0.2 · Minor Revisions

The connection between setae morphology and foraging ecology should be better explained. There are some issues related to formatting, terminology, and table clarity.

·

Basic reporting

Thanks for all the hard work you've put into revising the manuscript. It's clear you've made a strong effort to address the previous comments, and the manuscript is definitely improving. That said, there are still a few key points that could use more attention—especially the connection between setae morphology and foraging ecology, as well as providing clearer information on sample sizes in the Methods. I’ve also noted a few smaller issues related to formatting, terminology, and table clarity. I hope the comments below will be helpful as you continue refining your work—you're on the right track, and it's shaping up well.

Experimental design

no comment

Validity of the findings

no comment

Additional comments

Discussion: The revised sentence, “In stingless bees, the morphological diversity of the setae and structures involved in pollen handling of the hind tibiae and basitarsi are poorly investigated, despite the ecological and functional importance of these features and their potential to reveal evolutionary divergence among species with different foraging strategies (e.g., floral pollen collection, kleptoparasitism and necrophagy),” does not fully address the reviewer's original request. While it was placed in the first paragraph of the Discussion section, it seems to imply that the ecological and functional relevance of these features will be elaborated elsewhere—yet this discussion remains limited or unclear.

From lines 113–115: “The hind legs of Meliponini exhibit remarkable structural adaptations that allow for a variety of specialized functions, from pollen collection to transporting resin, seeds, mud, and feces used for nest construction (Grüter, 2020), as well as wing grooming (Michener, 2007),” it is unclear whether these cited studies directly investigate species with different foraging strategies or whether these species overlap with your focal taxa. Please clarify this point. If your dataset includes species that vary in foraging strategy, a more detailed discussion of the relationship between setae morphology and foraging ecology would significantly strengthen the paper.

Introduction: The phrase “…The stingless bees (tribu Meliponini)…” appears to use the French word "tribu" instead of the English term "tribe." Please correct it accordingly.

Methods: In the revised version, it is unclear how many bees were used for each measurement. While the supplementary file Raw_data.xlsx presents data as means ± SD or ranges, the actual sample sizes are not provided. Please clearly state the number of individuals measured per species or per morphological trait in the Methods section, as this information is important for reproducibility and interpretation.

Tables S2–S3: These tables contain numerous abbreviations. To improve readability, please include the full forms of these abbreviations in the table captions or as footnotes.

References: In the citation Conde-Boytel R, Erickson EH, Carlson SD. 1989, the scientific name used in that reference is not italicized. Please correct this formatting error.

Formatting: Please review the formatting of several lines, such as lines 69, 75, and 90. Some may require merging with the previous paragraph, while others may need indentation to indicate the start of a new paragraph.

Reviewer 3 ·

Basic reporting

After the first round of revisions, the authors incorporated all the suggestions to improve the robustness and clarity of the paper. Therefore, I am satisfied with the current version of the manuscript.

Experimental design

No comment.

Validity of the findings

No comment.

Additional comments

After the first round of revisions, the authors incorporated all the suggestions to improve the robustness and clarity of the paper. Therefore, I am satisfied with the current version of the manuscript.

---

## Round 0.3 · accepted · Accept

The authors provided all necessary revisions and now the paper can be accepted for publication.